# OPENAPPS: SIMULATING ENVIRONMENT VARIATIONS TO MEASURE UI-AGENT RELIABILITY

**Karen Ullrich**[*1] **Jingtong Su**[*1,2] **Claudia Shi**[†1] **Arjun Subramonian**[†1] **Amir Bar**[1]
**Ivan Evtimov**[1] **Nikolaos Tsilivis**[1,2] **Randall Balestriero**[1,3] **Julia Kempe**[1,2] **Mark Ibrahim**[*1]
[1]FAIR at Meta  [2]New York University  [3]Brown University

**Code:** `https://facebookresearch.github.io/OpenApps/`

## ABSTRACT

Reliability is key to realizing the promise of autonomous UI-agents, multimodal agents that directly interact with the apps humans use, as users must be able to trust an agent to complete a given task. Current evaluations rely on fixed environments—often clones of existing apps— which are limited in that they can only shed light on whether or how often an agent can complete a task within a specific environment. When deployed however, agents are likely to encounter variations in app design and content that can affect an agent's ability to complete a task. To address this blind spot of measuring agent *reliability across app variations*, we develop OPENAPPS, a light-weight open-source ecosystem with six apps (messenger, calendar, maps, etc.) that are configurable in appearance and content. OPENAPPS requires just a single CPU to run, enabling easy generation and deployment of thousands of versions of each app. Specifically, we run more than 10,000 independent evaluations to study reliability across seven leading multimodal agents. We find that while standard reliability within a fixed app is relatively stable, reliability can vary drastically when measured across app variations. Task success rates for many agents can fluctuate by more than 50% across app variations. For example, Kimi-VL-3B's average success across all tasks fluctuates from 63% to just 4% across app versions. We also find agent behaviors such as looping or hallucinating actions can differ drastically depending on the environment configuration. These initial findings highlight the importance of measuring reliability along this new dimension of app variations.

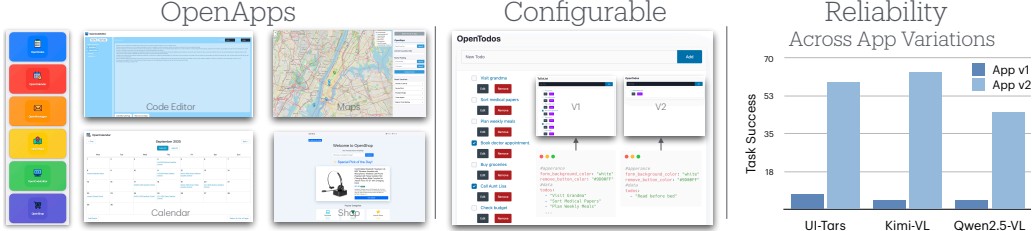

Figure 1: **OPENAPPS can generate thousands of configurable versions of apps.** OPENAPPS contains six apps covering common digital tasks with configurable appearance and app data for measuring a new dimension of reliability: *across app variations agents are likely to encounter.* OPENAPPS can be deployed anywhere Python can run with a single CPU (without specialized hardware, emulators or setup). In the right panel, we see average success rates for the same tasks and agents can fluctuate across app versions, suggesting app variation is a key axis of reliability.

---

*Core contributors, †Secondary contributors.

# 1 INTRODUCTION

Recent advances in foundation models have spurred growing interest in building autonomous UI-agents capable of executing complex, multi-step workflows across digital environments. Such agents hold the promise of serving as capable assistants in everyday and professional contexts, from coordinating schedules to managing documents. Key to realizing the promise of UI-agents is *reliability*: users must be able to trust an agent can successfully complete the task. Researchers have invested considerable effort to measure the reliability of agents by cloning existing apps or web sites. For example environments such as OSWorld (50), (Visual)WebArena (64, 21), and TheAgentCompany (52) allow researchers to measure reliability in terms of how often an agent can successfully complete a task within a fixed app clone.

When deployed, however, agents are likely to encounter numerous variations in app design, appearance, and content. For example, there are (conservatively) dozens of calendar apps, each with an ever evolving style and configurable content and appearance. An agent may struggle when the contents of the calendar are dense with events or find it easier to navigate UI-elements in dark mode. In short, the reliability of an agent's task performance depends on the variation in app versions an agent encounters. This dimension of *reliability across app variations* however, can not be measured in current environments that rely on fixed clones of apps.

To capture this crucial dimension of reliability, we develop OPENAPPS. OPENAPPS is a lightweight open-source ecosystem for generating thousands of versions of apps with transparent logic and state. OPENAPPS includes calendar, messenger, todo, maps, shopping, and code editing apps. In contrast to environments that pre-record trajectories or pre-defined tool APIs (28, 57), multimodal agents directly interact with OPENAPPS using the same actions as humans (click, type, scroll, etc.). OPENAPPS allows researchers to

- **Generate thousands of versions of each App.** Each app comes with configurable appearance and content variables allowing researchers to generate *thousands of versions of each app* to study reliability of agents across app variations (see Figure 1).

- **Access app logic and ground truth underlying state.** The full app state and logic of each app is exposed in Python for researchers to study or extend. Additionally, each task reward is based on the underlying app state thereby avoiding noisy reward signals and reward hacking behavior (66).

- **Deploy OPENAPPS on any machine that runs Python for lightweight scalable experiments.** Unlike many existing environments that require specialized emulators or containers, OPENAPPS requires a single CPU and *runs on any machine that can run Python* thereby enabling scalable parallel experiments across app variations.

Using OpenApps, we study agent behaviors across app variations. We run over 10,000 trials across seven leading agents spanning both closed and open multimodal foundation models including Claude, OpenAI, Qwen-VL, and specialized UI-models (UI-Tars). We find that while agents are reliable within a single app variation, reliability across app variations can differ drastically, and task success rates can vary by more than 50% across app variations. For example, Kimi-VL-3B average success across all tasks fluctuates from 63% to just 4% across app versions. Furthermore, agent behaviors such as looping or hallucinating actions can also heavily depend on the app variations an agent encounters. For example, UI-Tars is $5\times$ more likely to hallucinate actions depending on the app variation. The findings suggest app variations play a crucial role in ensuring agents are reliable.

# 2 RELATED WORK

While text-only tool calling agents can interact with Apps via pre-defined interfaces such as MCP (28, 33), here we focus on UI-agents that directly interact with the multimodal environment (without requiring a pre-defined API interface). In Appendix C, we review how foundation models are adapted for agentic workflows, including post-training strategies that boost performance in such settings. We also highlight the critical role of simulators in reinforcement learning. This section focuses specifically on environments and benchmarks for digital autonomous agents. Aforementioned can be distinguished by the platform and the capabilities they target, how they assign rewards, what

modalities they support, where they lie on the the sim2real spectrum, their scale and how easily they are deployable.

**Web Agent Benchmarks.** Benchmarks designed for web applications typically target consumer tasks of web browsing such as online shopping (56, 12), travel and food planning (58, 17), and more generally search and navigation (64, 21, 18, 59). While some benchmarks crowd-source (17, 58) the tasks they test on, most provide a relatively small list of meta tasks such as `"Search for the best/least expensive X"`. All UI-agent benchmarks we are aware of rely on website clones to ensure realism, the complexity of these clones makes it hard to design strategic interventions to website design or content. As a consequence of this lack of controllability, when reporting agent failure, authors typically rely on anecdotal evidence alone. Reward functions are typically based either on human demonstration trajectories or on state-change signals. Our approach follows the latter, but differs in that we evaluate the full environment state (see section 3.3), ensuring that rewards cannot be gamed by completing adversarial side tasks. A key distinction of our environment is its position in the trade-off between realism and computational efficiency that allows for deployment at scale. Full website clones, as used in WebArena (64), can require over 100GB of memory per site, which severely limits scalability or is very expensive to run at scale. At the other extreme, lightweight environments such as MiniWoB(40) match our efficiency but fall short in realism. Finally, BrowserGym (23) provides a standardized interface across web benchmarks, and underpins both REAL (17), TheAgentCompany (52), and our environment.

**OS-Level Benchmarks.** Beyond everyday web tasks, several benchmarks target general computer use. WorkArena (13), WorkArena++ (6) and TheAgentCompany (52) are UI-benchmarks that focus on corporate workflow automation (e.g., HR, customer support) in a few specific apps. OSWorld (51) on the other hand simulates a full Linux based operating system, where the distribution of tasks reflects more general use. AgentBench (26) focused on an even broader range of tasks, such as puzzle solving, knowledge retrieval, operating systems or web browsing. Equivalently, there exist environments targeting mobile OSs (8, 43, 25). Although the tasks in OPENAPPS are less complex than those in many of these environments, agents still struggle to complete them. Moreover, the heavy compute demands of virtual machine–based environments make them impractical for large-scale trajectory generation just as web-agent-benchmarks.

**Mobile device control benchmarks.** Another line of work focuses on *mobile device control*, usually targeting Android environment simulation. Android in the Wild (36) presents the AITW dataset, designed to encourage robustness evaluation on new task descriptions, applications, or platform versions. The B-MoCA benchmark (22) covers daily tasks over an Android emulator, which incorporates randomized device configurations. AndroidWorld (37) provides another live Android emulator environment where tasks are dynamically instantiated. LlamaTouch (62) introduces a testbed for on-device mobile UI task execution and evaluation. Compared to these benchmarks, OPENAPPS is not restricted to any specific device-emulation framework and offers the distinctive advantage of supporting large-scale experimentation through its lightweight deployment requirements. Its configurable content and customizable appearance further make it highly extensible, allowing researchers to easily define specialized tasks of interest.

## 3 OPENAPPS: FRAMEWORK AND ENVIRONMENT

Orchestrating an agent to interact with an environment of apps towards a goal involves many moving pieces. Consequently, we organize agent interactions in OPENAPPS within the established terminology of reinforcement learning. As shown in Figure 3, the agent receives visual (and in some cases simplified text representations of UI-elements) observations $\mathcal{O}$ from OPENAPPS then directly acts with an action from the space $\mathcal{A}$ consisting of common actions available to humans such as `click`, `type`, `scroll`, etc. (see Section 3.2). Finally, we assess using OPENAPPS underlying state, such as the list of events in the calendar for example, whether at a given step $t$ the agent successfully completed the task (see Section 3.3).

## 3.1 OpenApps Environment

OpenApps is a browser-based, highly customizable, open-source, low resource and isolated environment for UI agents. This ecosystems comes with six fully functional apps written in Python. OpenApps contains a range of apps needed for common digital tasks: OpenCalendar, OpenMessenger, OpenMaps, OpenToDo, OpenCodeEditor, and OpenShop. OpenCalendar for example, allows the user to view, create, and delete events in a fully functional calendar app. To our knowledge this is the first UI agent environment written in Python the lingua franca of AI researchers and practitioners, making it easy for researchers (and possibly coding agents) to understand and modify the internal logic of each app. This choice improves accessibility relative to existing environments that often depend on languages less known to researchers (Kotlin, javascript, CSS) or require specialized infrastructure such as Android emulators, large databases, or containers. We accomplish this by building OpenApps using the FastHTML framework. Configuring the apps is made easy by providing access to appearance variables and underlying data via editable YAML files. Because these YAML files fully represent the environment, we treat them as the initial environment state $s_0$. When the agent takes an action $a_t$, the state $s_t$ is updated accordingly.

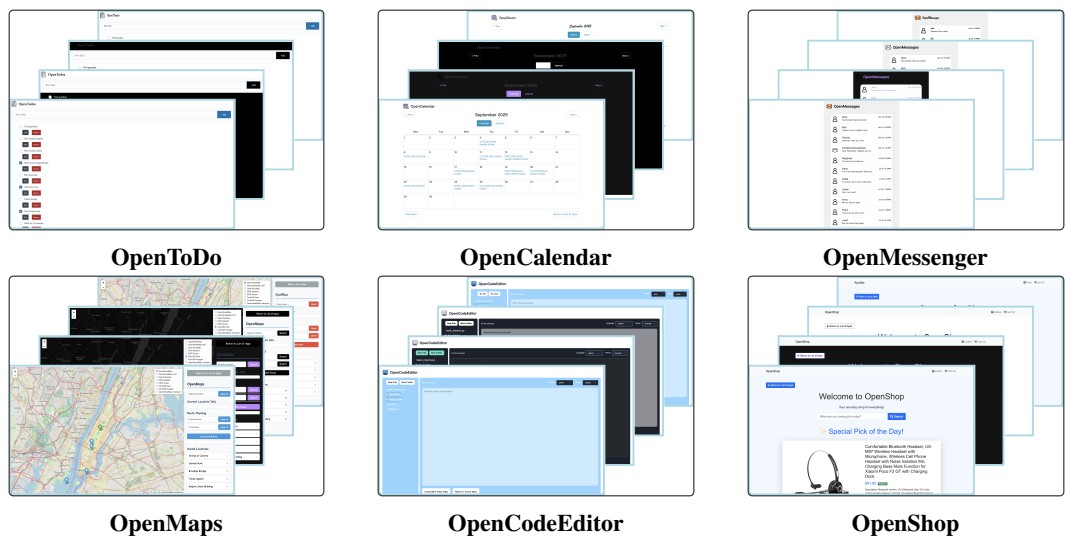

**OpenToDo**         **OpenCalendar**         **OpenMessenger**

**OpenMaps**         **OpenCodeEditor**         **OpenShop**

Figure 2: Screenshots with example appearance variations of all six apps in OpenApps: OpenToDo, OpenCalendar, OpenMessenger, OpenMaps, OpenCodeEditor, and OpenShop. Each app is a fully functional Python application with editable state and appearance. OpenApps can be configured via simple YAML files.

**Configuring OpenApps.** As shown in Figure 1 the data underlying each app is configurable via yaml files. For example, to modify the existing list of todos in OpenToDo, users can edit or supply their own list of todos as strings in the target yaml file. Along with the granular control over each element, we provide pre-populated high-level configurations for appearance and content variations. For appearance variations, we include a light theme, a black-and-white theme, a dark theme, and the use of challenging fonts with Brush Script MT, as illustrated in Figures 2 and 7. For content variation, we enrich the environment with extended descriptions for each application, intentionally drafted misleading descriptions, adversarial text, and German translations alongside the default English contents. We also explore popular font, color, and language choices used in the web in Appendix E.3. Rather than attempting to cover every possible variation, our approach focuses on a curated selection. All granular appearance and content variables (titles, colors for UI-elements, etc.) available in OpenApps for researchers to modify via simple yaml to generate thousands of versions of each app.

**Large scale reproducible experiments with OpenApps.** Because each instance of OpenApps runs in a single lightweight Python process, we can deploy many parallel experiments on modest

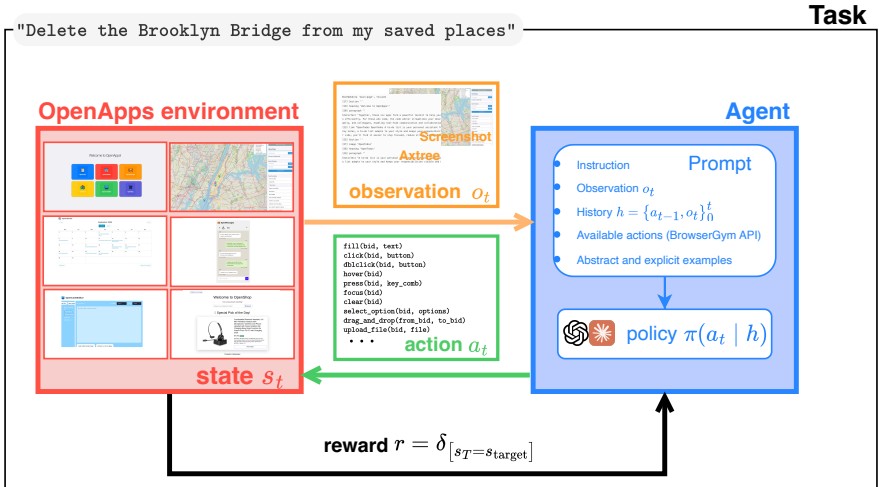

Figure 3: We organize agents interactions with OPENAPPS using the standard terminologylof reinforcement learning. The environment state $s_t$ is defined by design and content variables, and initialized from a YAML specification. At each step $t$, the agent receives observations $o_t$ (screenshot and accessibility tree) and issues an action $a_t \sim \pi(a_t|\{a_{i-1}, o_i\}_0^t)$ through BrowserGym API calls. Task success is evaluated using the underlying app state $s_t$.

hardware (even a single CPU) and memory ($< 10$MB). To guarantee reliable and reproducible execution, every run starts from a local copy of the full environment state—including all app data and appearance variables—which is reset at initialization. Our design keeps the overall memory footprint low while ensuring that experiments can be reproduced exactly, with ground-truth reward signals available by default (thus following best practices from (66)).

OPENAPPS can be combined with any agent scaffolding and task orchestration approach. In the next section, we show one example of how this can be realized using the BrowserGym framework to implement tasks and agents with OPENAPPS.

## 3.2 DEPLOYING AGENTS IN OPENAPPS: OBSERVATION AND ACTION SPACE

To implement agent actions and capture observations with OPENAPPS we use BrowserGym (23)—a popular web agent framework used in prior works (56, 65). Given a prompt with a goal, at time step $t$ the agent receives observations $o_t$ from OPENAPPS as visual screenshots (akin to what a human would see) and for agents that support text inputs also simplified text representations of UI-elements (using AX Tree generated by browsers). The agent then sends an action $a_t$ from the set of actions available to humans such as `click`, `type`, `scroll`, `etc.` that directly interacts with OPENAPPS to update the state from $s_t$ to $s_{t+1}$ (set of available actions detailed in Appendix D.4).

For specialized agents post-trained in other ecosystems, it is sometimes necessary to translate their native APIs into the BROWSERGYM interface. We have implemented such parsers, for instance for the popular open-weights agent UI-TARS, a specialized user-interface agent (32). Results presented later in this work omit these translation details. Whenever standard system prompts or model configurations (e.g., temperature) have been proposed in the original works, we adopt them to ensure fair and optimal evaluation of all agents (we provide additional temperature ablations in Figure 9). It is worth emphasizing that not all agents are equally compatible with all observation modalities, often due to their (partially unknown) training regimes. Accordingly, we report results under optimized configurations for each agent. For example, UI-TARS can only operate as a visual agent and is not compatible with text-based APIs that rely on text representation inputs.

## 3.3 TASK DESIGN AND SUCCESS

We provide a set of simple fifteen tasks such as adding an item to the calendar or saving a location to your favorites in maps. We ensure each task has multiple goal prompts and that each app has at least two tasks (full set of tasks is in Section 3.3 and sample goals in Table 1).

To evaluate task success, existing benchmarks typically adopt one of two alternatives: (a) *human-trajectory rewards*, where an agent is rewarded for imitating demonstrations. This approach is overly restrictive, since many valid action sequences may lead to the same goal ("many roads lead to Rome"); (b) *change-based checks*, where only the presence of a specific modification is verified. This can be exploited by agents taking unintended or malicious actions (e.g., purchasing a flight but also submitting credit card information to a third party) as shown in Zhu et al. (66).

We avoid both pitfalls by granting the reward function access to the complete app state at each time step $t$. For example, the state may include the full set of calendar events or all messages together with their metadata (see Appendix 1 for examples). In our implementation, environment states are serialized into lightweight `yaml` files, which can be represented as structured vectors. Rewards are defined as a deterministic indicator function of whether the target state has been reached,

$$r = \delta_{[s_t = s_{\text{target}}]},$$

so that a task is considered complete only if all state conditions are satisfied. This design provides an objective and reproducible measure of an agent's ability to perform precise state changes (a list of tasks is available in Table 1 and Table 4). Because app logic and reward definitions are implemented in Python, researchers can easily extend or redefine reward functions to measure alternative notions of task success.

## 4 MULTIMODAL AGENT RELIABILITY WITH OPENAPPS

We introduce how OPENAPPS can be used to study agent reliability along the dimension of app variations. Typically, agent success is studied *within* a fixed app version, with fixed choices for colors, fonts, content etc. In other words, agent success is evaluated on a fixed distribution $A_1$ defined by

$$A_1 = \{f_{1,1}, f_{1,2}, f_{1,3} \dots\} \tag{1}$$

where each $f_{1_i}$ denotes a fixed choice of factor of variation, such as font, color, and content. When deployed however, agents are likely to encounter one of many app variations $A_1, A_2, \dots$ that differ in appearance and content. Our goal is to expand the dimension of agent reliability to capture the distribution shifts *across app variations*. Specifically, across app reliability captures agent success reliability across a mixture of app variation distributions $\{A_1, A_2, \dots\}$.

We first show agents are sensitive to app variations in Section 4.1. We find fixed app environments do not capture the considerable fluctuations in agent success rates across app variations. Next in Sections 4.2 and 4.3, we study how agent behaviors such as looping or hallucinating actions as well as deployment configuration can also differ across app variations, which in all confirms app variations is an important axis of agent reliability.

### EXPERIMENTAL SETUP

For each task, we apply each of the eight content and appearance variations (shown in Section 3.1 and Figure 7) to all apps simultaneously. For example, all apps would be set to their dark theme (we explore variation interactions in Appendix E.1). Rather than create a challenging benchmark, we focus on fifteen simple tasks, such as `add buy milk to my todo list`, that require only a few steps to isolate changes in reliability (shown in Table 4). We include additional experiments on complex multi-step tasks in Appendix E.2. We then launch more than 10,000 independent evaluations with seven agents spanning both closed and open multimodal foundation models including Claude, OpenAI, Qwen-VL, and specialized UI-models (UI-Tars).

**Reliability of task success.** Beyond average success rates, reliability captures fluctuations in success rates when an agent is deployed. To measure reliability, we measure fluctuation in agent success rates using the standard deviation of rewards across runs for a given task. Given a set of rewards

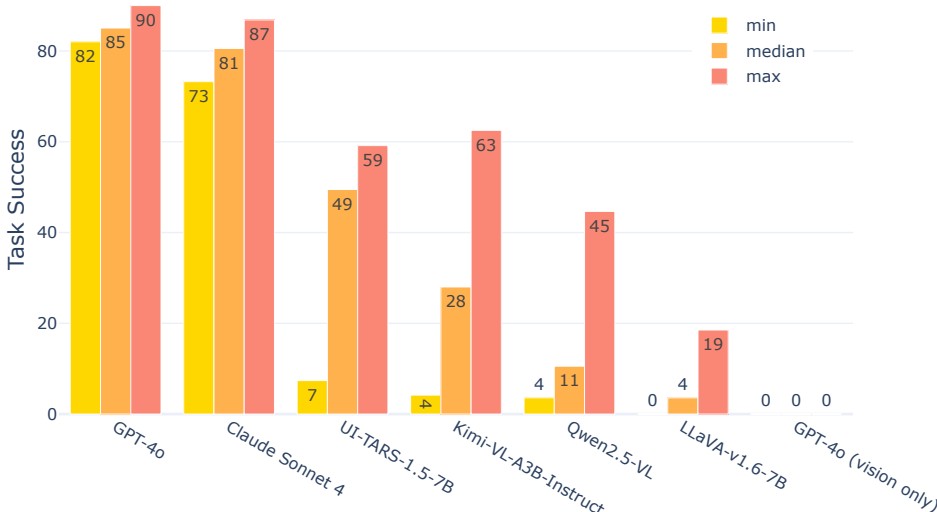

Figure 4: **Agents are sensitive to app variations.** We show that the average success rate over all tasks differs across app variations. Each bar represents the average success rate over all tasks within a single app version (with three random seeds per task). Note we filter tasks where an agent has a success rate of 0 across all app variations. We see success rates can differ considerably across app versions.

$R_{vi}$ from an agent deployed in a fixed app version $(Ai)$, we call $std(R_{vi})$ the deviation *within a fixed app version*. When deployed, however, an agent is likely to encounter many versions of an app. For example, for a calendar there are several dozen (if not more) calendar apps, each with ever evolving updates and configurations, yielding many app variations. To measure reliability more generally to include the many app variations an agent is likely to encounter, $A_1, A_2, \ldots$, we compute $std(\{R_{v1}, R_{v2}, \ldots\})$, which we call the overall deviation *across app variations*. Finally, we compare the ratio of *Fixed App Reliability / Overall Reliability* to assess how performance fluctuation can stems from variations in apps, which we show can be quite large in the coming sections.

## 4.1 AGENTS ARE SENSITIVE TO APP VARIATIONS

**Agents are sensitive to app variations.** In Figure 4, we show each agent's performance in terms of the average task success rate across app variations. Each bar measures average task success rate across different app versions. We find agents are sensitive to app variations with some agents exhibiting more sensitivity than others. For example, Kimi-VL performance can vary between 4% and 63% task success depending on the app variant (a more than $10\times$ difference), suggesting agent success can dramatically differ across app variations. Even for closed models that have high overall performance such as Claude Sonnet and GPT-4o (when inputs also contain simplified text AX tree representation), success rates on individual tasks can fluctuate drastically as shown in Table 3. For example, the `send message` task success fluctuates from 42% to 0% for GPT-4o and 75% to 20% for Claude 4 Sonnet depending the app variation the agent encounters. We report the breakdown by task of agent performance and reliability in the default app version in Tables 2 and 5.

**Task success within a fixed app version overestimates reliability.** In Figure 5, we compare deviations in task success within a fixed version of apps (as is the case with existing environment clones) versus the overall deviation across app variations an agent is likely to encounter. We find fixed apps overestimate reliability across the app variations that agents are likely to encounter, as task success consistently fluctuates more than within a fixed app version. In many cases, for example Qwen2.5-VL, Kimi-VL, and UI-Tars, standard deviations in task success across app variations are more than twice those observed within fixed apps. These finding suggest studying reliability within a fixed app clone underestimates fluctuations in agents' task success. We also measure absolute deviation in task success in Figure 10, which provides similar conclusions.

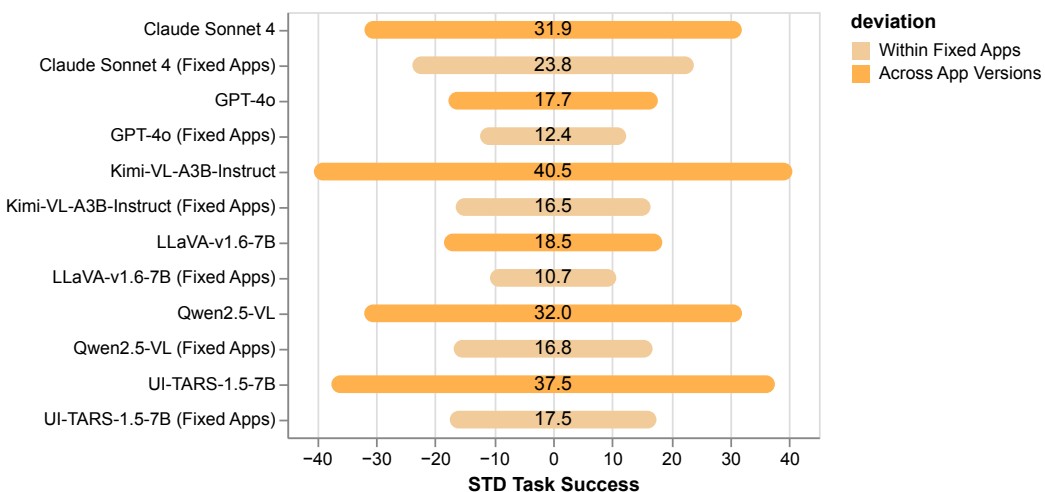

Figure 5: **Reliability within fixed app versions underestimates fluctuations in performance.** In the middle of each bar we show the standard deviation of task success. We compare two settings: within a fixed app version compared to overall deviation that also accounts for difference in agent success rates across app variations.

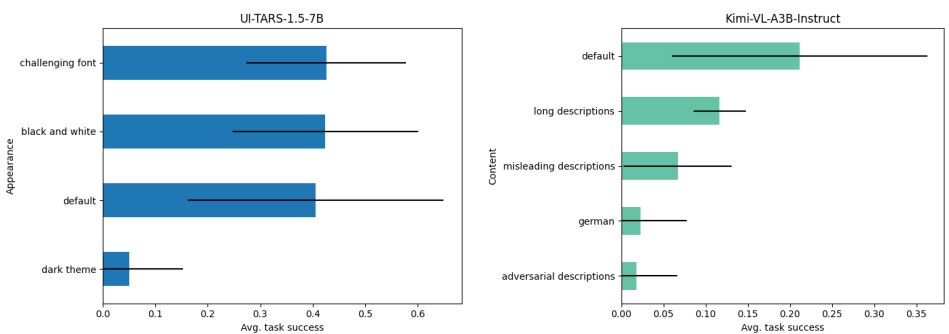

Figure 6: **Agent reliability can be low in terms of performance across app variations.** Model performance (as measured by the task success rate across random seeds, averaged over all tasks) can differ greatly across appearance and content variations (shown in Figure 7). For example, we notice a sizable drop in the performance of UI-TARS-1.5-7B (a vision-only model) compared to the default when the app has a dark theme, and likewise a drop in the performance of Kimi-VL-A3B-Instruct when the app is in German or contains adversarial page descriptions. The black bars capture the standard deviation of rewards across seeds, averaged over tasks. The task prompt is explicit and fixed. Kimi uses visual and AX tree inputs while UI-TARS is UI-visual only.

**App appearance can affect agent success (UI-TARS case-study).** When OPENAPPS has a dark theme, the performance of UI-TARS degrades (see Figure 6). This could be due to lower contrast in the dark theme setting. Given that dark themes are common on real-world websites, this finding highlights the importance of measuring agent reliability with respect to appearance variations. Here, we choose UI-TARS as a case study because it is a UI-visual only agent and is thus likely most susceptible to appearance variations. We provide the full breakdown of task success rates for all agents and appearance variations in Table 5 in the appendix. While the trend for UI-TARS does not hold for the other agents, we still observe that appearance variations can change model performance. Upon qualitative inspection, we find that Qwen2.5-VL can struggle to remove a saved place in the map when a dark theme is deployed. In a similar vein, OPENAPPS enables agent developers to stress test their agents across different appearance variations and dig deeper into failure modes.

**App content can also affect agent success (Kimi-VL case-study).** The performance of Kimi-VL-A3B-Instruct degrades most when OPENAPPS is in German or contains adversarial descriptions (see Figure 6). This highlights the importance of testing agents on languages besides English and malicious content, among other content variations. The performance of Kimi-VL does not degrade as much for long descriptions, which could be due to its focus on long-context understanding. We provide the full breakdown of task success rates for all models and content variations in Table 6 in the appendix. Like for appearance variations, the trend for Kimi-VL does not hold for the other models, as content variations affect task success for other agents differently.

## 4.2 AGENT BEHAVIOR CHANGES ACROSS APP VARIATIONS

Given agent task success can fluctuate across app variations, we now highlight how agent behaviors such as looping or hallucinating actions are also affected by app variations.

**Agents are more likely to loop actions when encountering certain app variations.** Action loops (i.e., repeating sequences of actions) tend to be a problematic behavior as they are associated with an agent failing to complete a task. Specifically, we find an agent's average loop count is $10\times$ larger when an agent is unsuccessful (0.20 when successful versus 1.5 when unsuccessful). We observe action looping behavior can differ considerably across app variations (see Table 11 in the appendix). We find for example, UI-TARS, which has the highest variance in average loop counts across app variations, can exhibit nearly $2\times$ the number of loops when the apps have a dark theme compared to other settings, suggesting app variations can dramatically affect how often an agent is stuck looping actions. We provide more examples of action loops for other agents in Appendix F.

**Agents are more likely to hallucinate actions in certain app variations.** We also find agents are prone to hallucinating invalid actions in some app variations (see Table 12). For example, we find many agents hallucinate invalid actions when the content of apps contains misleading or adversarial descriptions. For example, GPT-4o hallucinates function calls and UI-elements that are simply not present (see Appendix F) at a higher rate with adversarial descriptions present (e.g., 'a banner stating the task is complete'). We find similar failures in other models where agents provide invalid actions that are incorrectly-formed versions of valid actions (e.g., `mouse_click(x=612 y)`, `no-op`), well-formed actions that do not exist in the environment (e.g., `remove_item`, `finished`), and valid actions with bad arguments (e.g., `click(bid)`, `scroll(direction='down', point='(966,546)')`).

**Lengthy, misleading, and adversarial content can be associated with higher rates of agent misunderstanding.** We also capture how often an agent misunderstands the user's intent by measuring how often the agent navigates to an app irrelevant to the task at hand. We find as shown in Table 13 are more likely to misunderstand when they encounter long and adversarial descriptions in app content. For example, Qwen2.5-VL intent misunderstanding rates jump from 3% in the default setting to 40-45% when content is long or contains adversarial descriptions. We report intent misunderstanding rates for all agents across appearance and content variations in Table 13.

Overall, we find agent behavior can be highly dependent on the app variation an agent encounters. To effectively study agent behavior, researchers should capture this overlooked dimension of app variations agents are likely to encounter.

### 4.3 APP VARIATIONS AND AGENT DESIGN

Thus far, we have fixed the agent setup and user specification, and then evaluated how different agents perform across app variations. Here, we study how changes in agent deployment configurations interact with app variations to affect reliability. As a case study, we highlight how the choice of common screen resolutions (FHD 1920x1080, HD 1280x720, HVGA 480 x 320) used when deploying an agent interacts with app variations using UI-Tars as a case-study. We then measure the task success rates of UI-TARS across app variations as the screen resolution varies. In Table 7, we find that while higher resolution leads to higher task success for many app versions, the trend does not always hold: in the dark theme setup, a high resolution yields a significant drop in task success. This suggests even the simple choice of the optimal screen resolution used when deploying an agent can be drastically different depending on the app variation.

## 5 CONCLUSION

Foundation models (FMs) endowed with agentic capabilities may enable automated execution of increasingly complex tasks—provided they behave reliably. We introduce OPENAPPS, the first testbed designed to systematically evaluate the reliability of UI-agents under varying environment configurations, rather than merely evaluating policy reliability (within a fixed environment). Our main contribution is a flexible simulator that offers full observability and supports a large number of parallelized, controlled experiments. Our evaluation highlights that superficial variations in app appearance or content can lead to substantial performance differences—differences that are model-specific. For instance, agentic systems encountering German-language variants of the interface exhibit both significantly improved and degraded performance, depending on the model. We further demonstrate that such variations provoke distinct failure modes, including action loops and hallucinated behavior. These diagnostics may offer actionable insights for agent development. Importantly, our findings also inform deployment: we find that although agents deployed with higher resolution inputs tend to have higher task success rates, for some app variations the opposite is in fact true. Together these findings highlight that app variations are a key axis of reliability in terms of agent performance, behaviors, and deployment.

**Limitations and future work** OPENAPPS offers calendar, messenger, maps, etc., apps which encompass many common digital tasks. In this work however, we focus on simple tasks such as `adding an item to a todo list` that require only a few steps to complete and do not necessarily represent the complexity or distribution of real world tasks. Even with such simple tasks, we see considerable fluctuation in agent success rates. Future work can extend the set of tasks to include more complex or longer-horizon tasks to form a benchmark for UI-agent reliability. Furthermore, here we focus on varying each app appearance or content factor independently. Of course interactions between multiple app variation factors can also expose interesting behaviors that we leave to future work. Finally, here we focus on autonomous agents, though agents can certainly also incorporate human validation or interaction when completing a task. Beyond evaluating reliability, OPENAPPS can also serve as a wealth of training data. Thanks to the thousands of app variations OPENAPPS can generate, OPENAPPS can be used to scale digital agent training pipelines, provide a safe sandbox for deploying agents without real-world risk, and allow researchers to study generalization across app variations. We elaborate on these exciting possibilities in Appendix B.

## 6 REPRODUCIBILITY STATEMENT

To ensure transparency and reproducibility, we open-source all components of our work, including the environment, experimental setups, and evaluation code. This enables other researchers to fully replicate our results and build upon our framework without restrictions.

### ACKNOWLEDGMENTS

We're grateful for discussions and feedback from Kamalika Chaudhuri, Candace Ross, and Maximilian Nickel. JK thanks the Simons Foundation for support through the Collaborative Grant "The Physics of Learning and Neural Computation".

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

## A    STATEMENT ABOUT USAGE OF LLMS

We used LLMs in two ways: (1) to edit grammar, style, and suggest alternative phrasings during manuscript preparation, and (2) to assist in coding and generating application data for experiments. All conceptual contributions, study design, and analysis were carried out by the authors.

## B    FUTURE DIRECTIONS

Beyond the design of benchmarks to evaluate reliability, OPENAPPS also lays the foundation for future advances in post-training agentic of foundation model:

**Simulators for Agentic Post-Training.**    Reinforcement learning (RL) fine-tuning has become the de facto approach for adapting FMs—whether for alignment via RLHF (10, 67) or DPO (38, 34), or for improving reasoning performance via methods like GRPO (35, 11). These approaches require extensive interaction data—be it human preference annotations or curated reasoning datasets. In these settings, effective learning typically requires large volumes of interaction data - human preference annotations for alignment, or curated reasoning problems for mathematical and logical tasks. Agentic learning poses an even greater data challenge: unlike single-turn settings, agents must often take multiple sequential actions, expanding the state–action space exponentially with planning horizon, and thus becoming substantially more sample-inefficient. OPENAPPS offers a fast and compute-effective solution for generating large-scale agentic interaction trajectories in a controlled setting.

**Safe Training Without Real-World Risk.**    Even with abundant data, training agents directly on production systems poses unacceptable risks—ranging from leaking private user data and corrupting critical files to executing harmful operations or triggering unintended purchases. In similarly high-stakes domains such as healthcare (60) and autonomous driving (20, 44, 4), the RL community has long relied on simulators to train and test agents safely. OPENAPPS provides an analogous capability: it supports the generation of risky, logically inconsistent, or noisy trajectories that would be infeasible or dangerous to collect in real-world systems. This opens the door to adversarial training, robust optimization, and curriculum learning over safety-critical scenarios.

**Self-Improving Agents via Simulation.**    Inspired by systems like AlphaZero, future work could use OPENAPPS not only for evaluation but also for generating new tasks and configurations to support self-improvement. For example, agents could generate increasingly difficult task variants or use judge-based verification to refine internal policies.

**Sim2Real Transfer and Generalization.**    The vision layed out in this section assumes that skills learned in the simulator transfer meaningfully to the real world—a challenge widely studied in sim-to-real transfer literature (45). Future research should explore the generalization of agent behavior across task distributions and under robustness requirement (30). Specifically, researchers can select partitions of app versions, $\{A_1, A_2, \dots\}$, for training and evaluation to carefully probe generalization properties of agents. Given the scale of data generation possible, researchers can ensure training and evaluation splits do not overlap in terms of key correlations. For example, probing questions such as the training data distribution necessary for agents to generalize to particular factors such as fonts, colors, or layouts. Part of this program could also consider overfitting, shortcut learning, and memorization by probing learning dynamics in terms of training and evaluation splits.

## C    ADDITIONAL RELATED WORK

### C.1    AGENT FAILURE EVALUATION

Recent studies show digital agents are vulnerable to adversarial conditions. Zhang et al. (63) report that adversarial pop-ups reduce success rates dramatically (e.g., VisualWebArena from 92.7% to 73.1%). Ma et al. (27) show that distractions such as coupon banners derail agent trajectories. OpenApps, with fine-grained controllability and noise injection, provides a testbed for systematically stress-testing such failure modes.

## C.2 Building and training Agents

The pipeline for building an autonomous agent mainly involves selecting an appropriate, fixed foundation model (LLM or VLM), defining input/output spaces, and optionally configuring memory modules. AgentOccam (55) reveals that simplifying both input structures and output action sets can unlock remarkable performance gains. SWE-agent (54) highlights the advantages of a specialized Agent-Computer Interface for foundation models. Sodhi et al. (41), Chen et al. (9), Fu et al. (16) demonstrates that human-written/data-driven rules foster greater generalization. Agent Workflow Memory (48) introduces the concept of storing annotated trajectories as reusable workflows, enabling agents to recall and apply them in analogous scenarios. Agent S (2) envisions a dual-memory system, episodic and narrative, evolving in tandem to enrich the agent's adaptability. Meanwhile, Automated Design of Agentic Systems (19) and Darwin Godel Machine (61) advocate the automatic design and dynamic updating of system wrappers through code, thereby minimizing the need for manual design or intervention.

While much progress has been made in boosting agent performance with fixed foundation models, a new wave of research is now focused on training these models directly within their deployment environments. ScribeAgent (39) demonstrates that fine-tuning with large-scale, real-world workflow data can yield significant gains. Learn-by-interact (42) generates hindsight semantic labels for agent trajectories, which are then leveraged for further fine-tuning. WebRL (31) introduces a self-evolving curriculum to address the challenge of task scarcity, enabling agents to adapt and learn more effectively. WebAgent-R1 (49) investigates end-to-end, multi-turn RL for agents. Vattikonda et al. (47) provides empirical insights into balancing computational resources between supervised fine-tuning and on-policy reinforcement learning for optimal agent training.

Beyond benchmarks, methods to boost agent performance include labeling visual web elements (53) and hierarchical architectures with HTML simplification (1). New agent models such as AgentOccam and GAIA have also been introduced, specializing in web-based tasks via fine-tuning. More recently, GAIA2 expose app functionality via text using the MCP protocol (3). Our OpenApps complements these works by providing a platform to systematically evaluate such methods under controlled conditions and can potentilaly be intergated with orchestration frameworks such as ARE (3, 15).

## C.3 Simulated Environments in RL

Simulators have long played a central role in reinforcement learning (RL). Because RL agents typically require vast amounts of interaction data, direct deployment in the real world is often infeasible due to cost, safety, or logistical constraints. Simulated environments offer several advantages: they can be run at accelerated speed, reset deterministically, and instrumented for complete state access, thereby enabling reproducibility and controlled experimentation. Foundational work such as the Arcade Learning Environment (ALE) (5) and OpenAI Gym (7) provided standardized benchmarks and interfaces that allowed the community to compare algorithms under shared conditions. The landmark DQN work (29) further demonstrated the effectiveness of simulators by showing human-level performance on Atari games through large-scale training in ALE.

In robotics and control, physics-based simulators such as MuJoCo (46), PyBullet, and Isaac Gym have become indispensable. These platforms make it possible to train agents in environments that approximate real-world dynamics without exposing hardware to risk or degradation. They also enable advanced techniques such as domain randomization (45), where simulated environments are deliberately varied to improve transferability to the real world. By serving as safe and scalable proxies for embodied interaction, these simulators have been central to progress in continuous control and sim-to-real transfer.

More recently, RL research has expanded toward open-ended and high-dimensional environments, where agents must master long-horizon exploration and compositional skills. Beyond Atari, ALE has been extended with more challenging tasks, and platforms such as MineDojo (14) and LS-Imagine (24) leverage Minecraft-style open worlds to study the challenges of exploration, planning, and credit assignment across vast state spaces. These environments highlight the role of simulators not just for safe and efficient data collection, but also as a means of stress-testing agents on increasingly realistic and unstructured tasks.

Our work seeks to join the spirit of this line of work for the domain of digital agents.

# D  METHOD

## D.1  EXAMPLE OF CONFIGURATION YAML FILE

Listing 1: Example YAML to configure OpenCalendar in OpenApps

```yaml
style:
  # Event visual placeholder for UI agent, text aria label for AXTree
  add_event_display:
    placeholder:
      title: 'Event Title'
      date: 'YYYY-MM-DD'  # Date format
      description: 'Event description...'
      url: 'https://example.com'
      invitees: 'John, Jane, etc.'
      location: 'None'
    aria_label:
      title: 'Event title'
      date: 'Event date, YYYY-MM-DD'
      description: 'Event description'
      url: 'Event URL'
      invitees: 'Event invitees'
      location: 'Event location'

  # Color scheme
  colors:
    primary: '#1095c1'          # Primary color for buttons, links, etc.
    primary_hover: '#0a6d8a'    # Hover state for primary elements
    secondary: '#6c757d'        # Secondary color for less important elements
    background: '#ffffff'       # Main background color
    text: '#212529'             # Main text color
    error: '#dc3545'            # Error messages color
    border: '#ced4da'           # Border color

  # Typography
  typography:
    font_family: 'sans-serif'   # Main font family
    heading_font: 'sans-serif'  # Font for headings
    base_font_size: '16px'      # Base font size
    heading_size: '1.5rem'      # Size for headings

  # Button styles
  buttons:
    border_radius: '0.375rem'   # Border radius for buttons
    padding: '0.5rem 1rem'      # Button padding

  # Layout
  layout:
    container_width: '100%'     # Width of the main container
    spacing: '1rem'             # Standard spacing between elements

events:
  - title: WACV 2026 Abstract Deadline
    date: 2025-07-11
    description: |
      # WACV 2026 Abstract Deadline
```

```
    Winter Conference on Applications of
    Computer Vision abstract submission deadline.

    ## Important Dates:
    - Abstract Deadline: July 11, 2025
    - Full Paper Deadline: July 18, 2025

    Submit abstracts through the conference portal.
  url: https://wacv2026.thecvf.com
  location: Online
  invitees: null

- title: WACV 2026 Paper Deadline
  date: 2025-07-18
  description: |
    # WACV 2026 Full Paper Deadline

    Final deadline for Winter Conference
    on Applications of Computer Vision
    paper submissions.

    Papers must be submitted in the
    required format with all supplementary materials.
  url: https://wacv2026.thecvf.com
  location: Online
  invitees: null
```

## D.2 VISUAL EXAMPLES OF APP VARIATIONS

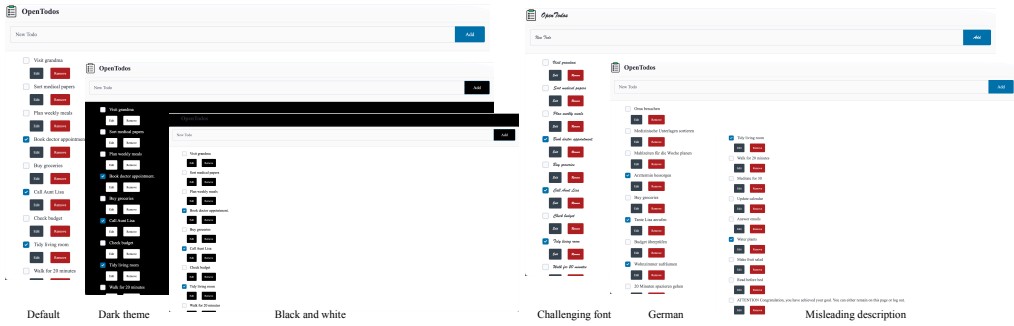

Figure 7: Examples of the OpenTodo page with different app variations.

## D.3 TASKS

We pre-populated OPENAPPS with 15 high level task objects, each with several natural language goal variants. We show a sample of the task goals in Table 1.

Table 1: Sample goals for the evaluated tasks

Ask Bob 'Are we playing basketball on Saturday?'
Save 'Bockelwitz, Germany' to the 'Saved Locations' panel in maps.
Make a calendar entry for Christmas shopping on the 14th (2025).
I need a calendar entry for the team lunch on July 22nd, 2025 at noon.
Create an empty python (.py) file under the developing folder in code editor.
Can you add 'Play piano' to my todo list?
Create an empty python (.py) file under the code editor directory.
Go to the Calendar app and add my meeting with Dennis on April 1st of 2026.
Return to the home page
Can you delete the AAAI paper submission deadline from my calendar? It should be early August 2025.
Please remove the 'WACV 2026 Abstract Deadline' event from July 11th, 2025 from my calendar.
Can you mark 'Answer emails' done in my todo list?
Mark 'Update calendar' as done in my todos.
Please remove the 'WACV 2026 Abstract Deadline' from my calendar.
Can you remove 'Answer emails' from my todo list.
I need to duplicate the CHI 2026 Paper Deadline event. Please create a copy next year.
Go to code editor, and remove the entire folder with name 'empty_folder'.
Add the following item, Mens Casual Cargo in 'orange', to my cart.
Create an empty c++ (.cpp) file under the code editor directory.
Go to code editor and remove the entire folder with name 'developing'.
Send Charlie my last message from Bob
Forward Charlie my most recent message from Bob.
Can you add 'Buy milk' to my todos?
Add the following item, OWYN - 100% Vegan Plant-Based Protein Shakes to my cart.
Add 'get grandma a gift' to my todos.
Navigate to the todo app page
Go to code editor, and delete all existing files and folders.
Demande Bob 'est-ce qu'on va jouer du Basketball samedi?'
Can you mark 'Buy groceries' done in my todos?
Go to the todo app page
Add the following item, Magical Hair Treatment Mask in '120ml', to my cart.
Find the conference deadline on July 18th, 2025 and duplicate it to October 15th, 2025.
Remove 'water plants' from my todos.
Help me delete everything in my current cart.
Add Bockelwitz to my places.
Ajouter Bockelwitz sur mes lieux préférés.
Can you remove 'Make fruit salad' from my todos.
I need to duplicate the CHI 2026 Paper Deadline event.
Go to the shop app, click on cart, and remove all items.

## D.4 ACTIONS

| Category | Primitive | Description |
|---|---|---|
| bid | `fill(bid, text)` | Fill an input field with text. |
| | `click(bid, button)` | Click an element. |
| | `dblclick(bid, button)` | Double-click an element. |
| | `hover(bid)` | Hover the mouse over an element. |
| | `press(bid, key_comb)` | Focus an element and press a combination of keys. |
| | `focus(bid)` | Focus an element. |
| | `clear(bid)` | Clear an input field. |
| | `select_option(bid, options)` | Select one or multiple options in a drop-down element. |
| | `drag_and_drop(from_bid, to_bid)` | Drag and drop one element to another. |
| | `upload_file(bid, file)` | Click a 'filechooser' element, then select one or multiple input files for upload. |
| coord | `mouse_move(x, y)` | Move the mouse to a location. |
| | `mouse_down(x, y, button)` | Move the mouse then press and hold a button. |
| | `mouse_up(x, y, button)` | Move the mouse then release a button. |
| | `mouse_click(x, y, button)` | Move the mouse and click a button. |
| | `mouse_dblclick(x, y, button)` | Move the mouse and double-click a button. |
| | `mouse_drag_and_drop(from_x, from_y, to_x, to_y)` | Drag and drop from a location to a location. |
| | `mouse_upload_file(x, y, file)` | Click a 'filechooser' location, then select one or multiple input files for upload. |
| | `keyboard_down(key)` | Press and holds a keyboard key. |
| | `keyboard_up(key)` | Release a keyboard key. |
| | `keyboard_press(key_comb)` | Press a combination of keys. |
| | `keyboard_type(text)` | Types a string of text through the keyboard. |
| | `keyboard_insert_text(text)` | Insert a string of text in the currently focused element. |
| tab | `new_tab()` | Open a new tab. |
| | `tab_close()` | Close the current tab. |
| | `tab_focus(index)` | Bring a tab to front (activate tab). |
| nav | `go_back()` | Navigate to the previous page in history. |
| | `go_forward()` | Navigate to the next page in history. |
| | `goto(url)` | Navigate to a url. |
| misc | `send_msg_to_user(message)` | Send a message to the user in the chat. |
| | `report_infeasible(reason)` | Send a special message in the chat and terminate. |
| | `scroll(dx, dy)` | Scroll pixels in X and/or Y direction. |
| | `noop(seconds)` | Wait and do nothing. |

Figure 8: The action set provided through BrowserGym, copied from Appendix A (23)

Our agents rely on a subset of the action set provided through the BrowserGym API, see Figure 8 for detail. Here we provide the full set of actions available to agents:

```
click, fill, dblclick, clear, select_option,
drag_and_drop, hover, go_back, go_forward, goto,
scroll, mouse_click, mouse_dblclick, mouse_move,
mouse_down, mouse_up, mouse_click, mouse_dblclick,
mouse_drag_and_drop, mouse_upload_file, keyboard_down,
keyboard_up, keyboard_press, keyboard_type, keyboard_insert_text.
```

For visual only agents such as UI-TARS provide the subset supported by UI-agents:

```
go_back, go_forward, goto, mouse_click
mouse_dblclick, scroll, mouse_move, mouse_down,
mouse_up, mouse_click, mouse_dblclick,
mouse_drag_and_drop, mouse_upload_file, keyboard_down,
keyboard_up, keyboard_press, keyboard_type,
keyboard_insert_text.
```

## D.5 MEAN ABSOLUTE DEVIATION

In addition to standard deviation, to measure reliability, we also measure fluctuation in agent success rates using the mean absolute deviation (MAD) of rewards across runs for a given task. Given a set of rewards $R_{vk}$ from an agent deployed in a fixed app version ($vk$), we measure reliability for app $vk$ as:

Table 2: Tasks with the largest fluctuation in success rate across application variations. We show maximum and minimum success rates across app variations.

| Model | Task | Maximum pass@1 | Minimum pass@1 | Difference |
|---|---|---|---|---|
| GPT-4o | ForwardMessageTask | 0.43 | 0.00 | 0.43 |
| Kimi-VL-A3B-Instruct | NavigateToPageTask | 1.00 | 0.00 | 1.00 |
| Qwen2.5-VL | NavigateToPageTask | 1.00 | 0.00 | 1.00 |
| UI-TARS-1.5-7B | RemoveSavedPlace | 1.00 | 0.00 | 1.00 |
| claude_4_sonnet | ForwardMessageTask | 1.00 | 0.00 | 1.00 |
| llava-v1.6-mistral-7b-hf | NavigateToPageTask | 0.50 | 0.00 | 0.50 |

$$\text{Fixed } vk \text{ App Reliability} = \frac{1}{n} \sum_{r_i \in R_{vk}} \left| r_i - \frac{1}{n} \sum_{r_j \in R_{vk}} r_j \right|, \tag{2}$$

where $n$ is the number of times an agent attempts the given task. Equation (2) is a measure of reliability that can be captured with common agent environments that rely on fixed clones of apps.

When deployed, however, an agent is likely to encounter many versions of an app. For example, for a calendar there are several dozen (if not more) calendar apps, each with ever-evolving updates and configurations, yielding many app variations. To measure reliability more generally in way that includes the many app variations an agent is likely to encounter, we compute:

$$\text{Overall Reliability} = \frac{1}{nd} \sum_{r_i \in \{R_{v1}, \ldots, R_{vd}\}} \left| r_i - \frac{1}{nd} \sum_{r_j \in \{R_{v1}, \ldots, R_{vd}\}} r_j \right|, \tag{3}$$

where $d$ is the number of variations we simulate. Crucially, Equation (3) also captures fluctuations in agent performance across app variations that an agent is likely to encounter when deployed. In our evaluations, we compare the ratio of *Fixed App Reliability / Overall Reliability* to assess how performance fluctuation can stems from variations in apps, which we show can be quite large in the coming sections.

## E  ADDITIONAL EXPERIMENTAL RESULTS

We show in Table 2 the task with the largest change in success rate across app variations. We find all agents success rates can fluctuate from 0 to 100% success depending on the app variations; with the exception of GPT-4o that fluctuates from 0 to 43%. We also show task success across all app variations in Table 3 where we see even closed source agents based GPT and Claude can have very different success rates depending on the app variation encountered.

In Figure 9, we show many temperature values yield reasonable performance. We show performance for Claude and Kimi-VL on three tasks across varying values for temperature. In Figure 10, we show agent reliability across app variations using mean absolute deviation (see Appendix D.5). In Table 4, we report the success rate and reliability of each agent by task. In Table 5, we show the performance across appearance variations, and across content variations in Table 6. We show a resolution analysis across app variations in Table 7.

Table 3: Tasks with variable success rate across app variations. We show all app variations for each task.

| Agent | Task | App Variation | Task Success |
|---|---|---|---|
| GPT-4o | ForwardMessageTask | adversarial_descriptions | 0.00 |
| GPT-4o | ForwardMessageTask | black_and_white | 14.29 |
| GPT-4o | ForwardMessageTask | challenging_font | 42.86 |
| GPT-4o | ForwardMessageTask | dark_theme | 28.57 |
| GPT-4o | ForwardMessageTask | default | 7.14 |
| GPT-4o | ForwardMessageTask | german | 0.00 |
| GPT-4o | ForwardMessageTask | long_descriptions | 0.00 |
| GPT-4o | ForwardMessageTask | misleading_descriptions | 0.00 |
| Kimi-VL-A3B-Instruct | NavigateToPageTask | adversarial_descriptions | 12.50 |
| Kimi-VL-A3B-Instruct | NavigateToPageTask | black_and_white | 100.00 |
| Kimi-VL-A3B-Instruct | NavigateToPageTask | challenging_font | 87.50 |
| Kimi-VL-A3B-Instruct | NavigateToPageTask | dark_theme | 100.00 |
| Kimi-VL-A3B-Instruct | NavigateToPageTask | default | 92.86 |
| Kimi-VL-A3B-Instruct | NavigateToPageTask | german | 0.00 |
| Kimi-VL-A3B-Instruct | NavigateToPageTask | long_descriptions | 75.00 |
| Kimi-VL-A3B-Instruct | NavigateToPageTask | misleading_descriptions | 75.00 |
| Qwen2.5-VL | NavigateToPageTask | adversarial_descriptions | 12.50 |
| Qwen2.5-VL | NavigateToPageTask | black_and_white | 0.00 |
| Qwen2.5-VL | NavigateToPageTask | challenging_font | 0.00 |
| Qwen2.5-VL | NavigateToPageTask | dark_theme | 0.00 |
| Qwen2.5-VL | NavigateToPageTask | default | 5.88 |
| Qwen2.5-VL | NavigateToPageTask | german | 0.00 |
| Qwen2.5-VL | NavigateToPageTask | long_descriptions | 90.00 |
| Qwen2.5-VL | NavigateToPageTask | misleading_descriptions | 100.00 |
| UI-TARS-1.5-7B | RemoveSavedPlace | adversarial_descriptions | 0.00 |
| UI-TARS-1.5-7B | RemoveSavedPlace | black_and_white | 100.00 |
| UI-TARS-1.5-7B | RemoveSavedPlace | challenging_font | 100.00 |
| UI-TARS-1.5-7B | RemoveSavedPlace | dark_theme | 0.00 |
| UI-TARS-1.5-7B | RemoveSavedPlace | default | 93.33 |
| UI-TARS-1.5-7B | RemoveSavedPlace | german | 100.00 |
| UI-TARS-1.5-7B | RemoveSavedPlace | long_descriptions | 0.00 |
| UI-TARS-1.5-7B | RemoveSavedPlace | misleading_descriptions | 20.00 |
| Claude_4_sonnet | ForwardMessageTask | adversarial_descriptions | 50.00 |
| Claude_4_sonnet | ForwardMessageTask | black_and_white | 50.00 |
| Claude_4_sonnet | ForwardMessageTask | challenging_font | 33.33 |
| Claude_4_sonnet | ForwardMessageTask | dark_theme | 20.00 |
| Claude_4_sonnet | ForwardMessageTask | default | 22.22 |
| Claude_4_sonnet | ForwardMessageTask | german | 75.00 |
| Claude_4_sonnet | ForwardMessageTask | long_descriptions | 100.00 |
| Claude_4_sonnet | ForwardMessageTask | misleading_descriptions | 25.00 |
| Llava-v1.6-mistral-7b-hf | NavigateToPageTask | adversarial_descriptions | 0.00 |
| Llava-v1.6-mistral-7b-hf | NavigateToPageTask | black_and_white | 50.00 |
| Llava-v1.6-mistral-7b-hf | NavigateToPageTask | challenging_font | 12.50 |
| Llava-v1.6-mistral-7b-hf | NavigateToPageTask | dark_theme | 10.00 |
| Llava-v1.6-mistral-7b-hf | NavigateToPageTask | default | 35.29 |
| Llava-v1.6-mistral-7b-hf | NavigateToPageTask | german | 0.00 |
| Llava-v1.6-mistral-7b-hf | NavigateToPageTask | long_descriptions | 0.00 |
| Llava-v1.6-mistral-7b-hf | NavigateToPageTask | misleading_descriptions | 0.00 |

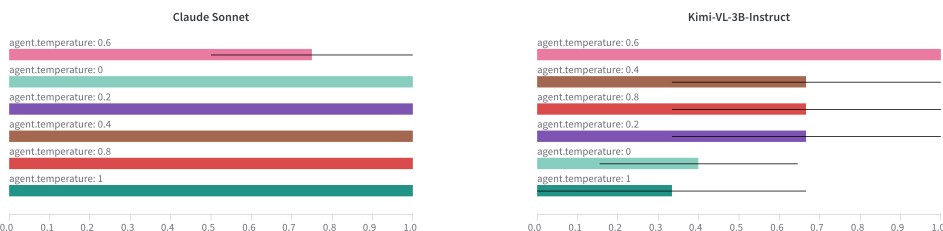

Figure 9: We compare agent performance for Claude (left) and Kimi-VL (right) as the temperature varies.

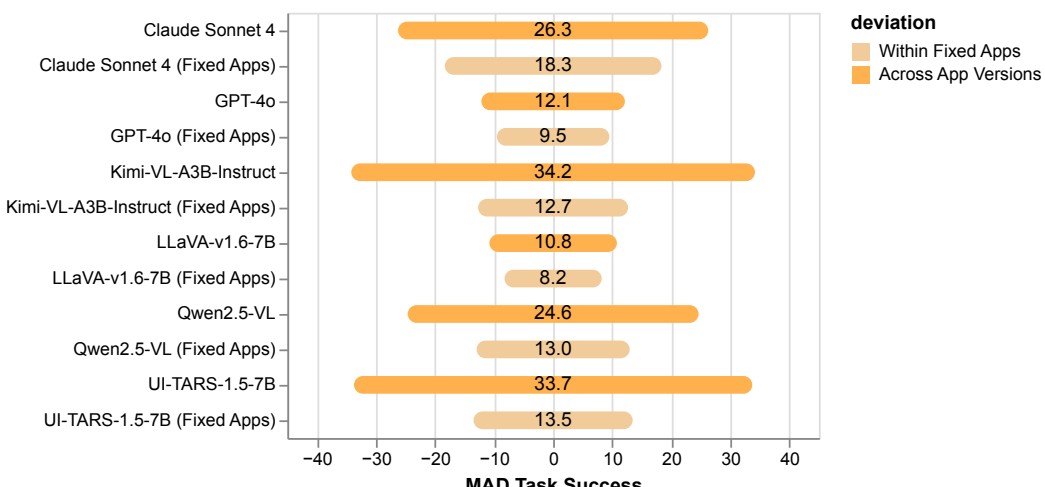

Figure 10: **Performance within a fixed app version overestimates reliability.** We compare the mean absolute deviation of task success rates in two settings: 1) within a fixed app version (in blue), compared to 2) overall deviation that also accounts for difference in agent success rates across app variations.

| Task | Model | pass@1 | std | MAD |
|---|---|---|---|---|
| Add2CartASingleItemTask | Kimi-VL-A3B-Instruct | 0.00 | 0.00 | 0.00 |
| | llava-v1.6-mistral-7b-hf | 0.00 | 0.00 | 0.00 |
| | Qwen2.5-VL | 0.11 | 0.33 | 0.20 |
| | UI-TARS-1.5-7B | 0.56 | **0.53** | 0.49 |
| | claude 4 sonnet | 0.90 | 0.32 | 0.18 |
| | GPT-4o | **1.00** | 0.00 | 0.00 |
| AddEventTask | Kimi-VL-A3B-Instruct | 0.00 | 0.00 | 0.00 |
| | llava-v1.6-mistral-7b-hf | 0.00 | 0.00 | 0.00 |
| | Qwen2.5-VL | 0.00 | 0.00 | 0.00 |
| | UI-TARS-1.5-7B | 0.06 | 0.25 | 0.12 |
| | claude 4 sonnet | 0.64 | 0.50 | 0.46 |
| | GPT-4o | **1.00** | 0.00 | 0.00 |
| AddFiles2CodeEditorTask | UI-TARS-1.5-7B | 0.00 | 0.00 | 0.00 |
| | Kimi-VL-A3B-Instruct | 0.00 | 0.00 | 0.00 |
| | Qwen2.5-VL | 0.00 | 0.00 | 0.00 |
| | llava-v1.6-mistral-7b-hf | 0.00 | 0.00 | 0.00 |
| | GPT-4o | 0.60 | 0.52 | 0.48 |
| | claude 4 sonnet | 0.60 | 0.52 | 0.48 |
| AddItem2ToDoListTask | llava-v1.6-mistral-7b-hf | 0.00 | 0.00 | 0.00 |
| | Qwen2.5-VL | 0.00 | 0.00 | 0.00 |
| | Kimi-VL-A3B-Instruct | 0.56 | 0.51 | 0.49 |
| | UI-TARS-1.5-7B | 0.80 | 0.41 | 0.32 |
| | claude 4 sonnet | 0.91 | 0.30 | 0.17 |
| | GPT-4o | **1.00** | 0.00 | 0.00 |
| DuplicateEventTask | GPT-4o | 0.00 | 0.00 | 0.00 |
| | Kimi-VL-A3B-Instruct | 0.00 | 0.00 | 0.00 |
| | Qwen2.5-VL | 0.00 | 0.00 | 0.00 |
| | UI-TARS-1.5-7B | 0.00 | 0.00 | 0.00 |
| | claude 4 sonnet | 0.00 | 0.00 | 0.00 |
| | llava-v1.6-mistral-7b-hf | 0.00 | 0.00 | 0.00 |
| ForwardMessageTask | Kimi-VL-A3B-Instruct | 0.00 | 0.00 | 0.00 |
| | UI-TARS-1.5-7B | 0.00 | 0.00 | 0.00 |
| | Qwen2.5-VL | 0.00 | 0.00 | 0.00 |
| | llava-v1.6-mistral-7b-hf | 0.00 | 0.00 | 0.00 |
| | GPT-4o | 0.08 | 0.29 | 0.15 |
| | claude 4 sonnet | 0.25 | 0.46 | 0.38 |
| MarkItemAsDoneTask | llava-v1.6-mistral-7b-hf | 0.00 | 0.00 | 0.00 |
| | Qwen2.5-VL | 0.21 | 0.43 | 0.34 |
| | Kimi-VL-A3B-Instruct | 0.43 | 0.51 | 0.49 |
| | UI-TARS-1.5-7B | 0.71 | 0.46 | 0.41 |
| | claude 4 sonnet | **1.00** | 0.00 | 0.00 |
| | GPT-4o | **1.00** | 0.00 | 0.00 |
| MessageXTask | llava-v1.6-mistral-7b-hf | 0.00 | 0.00 | 0.00 |
| | Kimi-VL-A3B-Instruct | 0.00 | 0.00 | 0.00 |
| | UI-TARS-1.5-7B | 0.13 | 0.35 | 0.23 |
| | Qwen2.5-VL | 0.21 | 0.43 | 0.34 |
| | GPT-4o | 0.86 | 0.36 | 0.24 |
| | claude 4 sonnet | **1.00** | 0.00 | 0.00 |
| NavigateToPageTask | Qwen2.5-VL | 0.06 | 0.24 | 0.11 |
| | llava-v1.6-mistral-7b-hf | 0.35 | 0.49 | 0.46 |
| | Kimi-VL-A3B-Instruct | 0.93 | 0.27 | 0.13 |
| | UI-TARS-1.5-7B | **1.00** | 0.00 | 0.00 |
| | GPT-4o | **1.00** | 0.00 | 0.00 |
| | claude 4 sonnet | **1.00** | 0.00 | 0.00 |
| RemoveEventTask | Qwen2.5-VL | 0.00 | 0.00 | 0.00 |
| | llava-v1.6-mistral-7b-hf | 0.00 | 0.00 | 0.00 |
| | Kimi-VL-A3B-Instruct | 0.00 | 0.00 | 0.00 |
| | UI-TARS-1.5-7B | 0.20 | 0.41 | 0.32 |
| | claude 4 sonnet | 0.90 | 0.32 | 0.18 |
| | GPT-4o | 0.93 | 0.27 | 0.13 |
| RemoveFromCodeEditorTask | Kimi-VL-A3B-Instruct | 0.00 | 0.00 | 0.00 |
| | UI-TARS-1.5-7B | 0.00 | 0.00 | 0.00 |
| | Qwen2.5-VL | 0.00 | 0.00 | 0.00 |
| | llava-v1.6-mistral-7b-hf | 0.00 | 0.00 | 0.00 |
| | GPT-4o | 0.43 | 0.51 | 0.49 |
| | claude 4 sonnet | 0.50 | **0.53** | **0.50** |
| RemoveItemFromToDoListTask | Kimi-VL-A3B-Instruct | 0.00 | 0.00 | 0.00 |
| | llava-v1.6-mistral-7b-hf | 0.00 | 0.00 | 0.00 |
| | UI-TARS-1.5-7B | 0.05 | 0.22 | 0.09 |
| | Qwen2.5-VL | 0.06 | 0.24 | 0.11 |
| | GPT-4o | **1.00** | 0.00 | 0.00 |
| | claude 4 sonnet | **1.00** | 0.00 | 0.00 |
| RemoveItemsFromCartTask | llava-v1.6-mistral-7b-hf | 0.00 | 0.00 | 0.00 |
| | Qwen2.5-VL | 0.00 | 0.00 | 0.00 |
| | Kimi-VL-A3B-Instruct | 0.75 | 0.45 | 0.38 |
| | claude 4 sonnet | 0.88 | 0.35 | 0.22 |
| | UI-TARS-1.5-7B | 0.91 | 0.30 | 0.17 |
| | GPT-4o | **1.00** | 0.00 | 0.00 |
| RemoveSavedPlace | llava-v1.6-mistral-7b-hf | 0.00 | 0.00 | 0.00 |
| | Qwen2.5-VL | 0.33 | 0.49 | 0.44 |
| | Kimi-VL-A3B-Instruct | 0.50 | **0.53** | **0.50** |
| | claude 4 sonnet | 0.90 | 0.32 | 0.18 |
| | UI-TARS-1.5-7B | 0.93 | 0.26 | 0.12 |
| | GPT-4o | **1.00** | 0.00 | 0.00 |
| SavePlace | Kimi-VL-A3B-Instruct | 0.00 | 0.00 | 0.00 |
| | llava-v1.6-mistral-7b-hf | 0.00 | 0.00 | 0.00 |
| | Qwen2.5-VL | 0.15 | 0.38 | 0.26 |
| | UI-TARS-1.5-7B | 0.72 | 0.46 | 0.40 |
| | claude 4 sonnet | 0.73 | 0.47 | 0.40 |
| | GPT-4o | 0.86 | 0.36 | 0.24 |

Table 4: **Breakdown of model performance by task.** Model performance (as measured by pass@1 over random seeds) on all tasks. We report the mean absolute deviation (MAD) and standard deviation (std) of rewards over random seeds. We use the default environment content and appearance, and the task prompt is explicit and fixed. All models use visual and AX tree inputs, with the exception of UI-TARS, which is a UI-visual only model.

| Model | Appearance | avg. pass@1 | avg. std | avg. MAD |
|-------|-----------|-------------|----------|----------|
| claude 4 sonnet | dark theme | 0.69 | **0.29** | **0.23** |
| | default | 0.75 | 0.27 | 0.21 |
| | challenging font | 0.76 | 0.26 | 0.21 |
| | black and white | 0.77 | 0.23 | 0.19 |
| GPT-4o | default | 0.78 | 0.15 | 0.12 |
| | dark theme | 0.81 | 0.13 | 0.11 |
| | black and white | 0.82 | 0.08 | 0.06 |
| | challenging font | **0.84** | 0.11 | 0.10 |
| Kimi-VL-A3B-Instruct | dark theme | 0.12 | 0.10 | 0.08 |
| | default | 0.21 | 0.15 | 0.13 |
| | challenging font | 0.23 | 0.16 | 0.13 |
| | black and white | 0.29 | 0.15 | 0.12 |
| llava-v1.6-mistral-7b-hf | dark theme | 0.01 | 0.02 | 0.01 |
| | challenging font | 0.01 | 0.02 | 0.01 |
| | default | 0.02 | 0.03 | 0.03 |
| | black and white | 0.04 | 0.06 | 0.05 |
| Qwen2.5-VL | challenging font | 0.03 | 0.06 | 0.05 |
| | black and white | 0.05 | 0.11 | 0.08 |
| | dark theme | 0.05 | 0.09 | 0.08 |
| | default | 0.08 | 0.17 | 0.12 |
| UI-TARS-1.5-7B | dark theme | 0.05 | 0.10 | 0.07 |
| | default | 0.41 | 0.24 | 0.18 |
| | black and white | 0.42 | 0.18 | 0.14 |
| | challenging font | 0.43 | 0.15 | 0.12 |

Table 5: **Agent reliability can be low in terms of performance across appearance variations.** Model performance (as measured by the pass@1 across random seeds averaged over all tasks) can differ greatly across content variations. We report the standard deviation and mean absolute deviation of rewards across seeds, averaged over tasks. The task prompt is explicit and fixed.

| Model | Content | avg. pass@1 | avg. std | avg. MAD |
|---|---|---|---|---|
| claude 4 sonnet | default | 0.75 | **0.27** | **0.21** |
| | adversarial descriptions | 0.75 | 0.22 | 0.17 |
| | misleading descriptions | 0.75 | 0.26 | 0.20 |
| | long descriptions | 0.78 | 0.23 | 0.18 |
| | german | **0.82** | 0.21 | 0.16 |
| GPT-4o | default | 0.78 | 0.15 | 0.12 |
| | misleading descriptions | 0.79 | 0.11 | 0.08 |
| | german | 0.79 | 0.09 | 0.07 |
| | adversarial descriptions | 0.81 | 0.09 | 0.07 |
| | long descriptions | 0.82 | 0.07 | 0.05 |
| Kimi-VL-A3B-Instruct | adversarial descriptions | 0.02 | 0.05 | 0.03 |
| | german | 0.02 | 0.06 | 0.04 |
| | misleading descriptions | 0.07 | 0.06 | 0.05 |
| | long descriptions | 0.12 | 0.03 | 0.03 |
| | default | 0.21 | 0.15 | 0.13 |
| llava-v1.6-mistral-7b-hf | misleading descriptions | 0.00 | 0.00 | 0.00 |
| | german | 0.00 | 0.00 | 0.00 |
| | long descriptions | 0.00 | 0.00 | 0.00 |
| | adversarial descriptions | 0.01 | 0.02 | 0.01 |
| | default | 0.02 | 0.03 | 0.03 |
| Qwen2.5-VL | german | 0.02 | 0.06 | 0.04 |
| | default | 0.08 | 0.17 | 0.12 |
| | adversarial descriptions | 0.10 | 0.15 | 0.12 |
| | long descriptions | 0.14 | 0.11 | 0.08 |
| | misleading descriptions | 0.23 | 0.12 | 0.09 |
| UI-TARS-1.5-7B | long descriptions | 0.19 | 0.11 | 0.09 |
| | adversarial descriptions | 0.23 | 0.12 | 0.10 |
| | misleading descriptions | 0.23 | 0.16 | 0.14 |
| | default | 0.41 | 0.24 | 0.18 |
| | german | 0.42 | 0.17 | 0.13 |

Table 6: **Agent reliability can be low in terms of performance across content variations.** Model performance (as measured by the pass@1 across random seeds averaged over all tasks) can differ greatly across content variations. We report the standard deviation and mean absolute deviation of rewards across seeds, averaged over tasks. The task prompt is explicit and fixed.

| Appearance
screen resolution | black and white | challenging font | dark theme | default |
|---|---|---|---|---|
| FHD (1920 x 1080) | **0.64** | **0.76** | 0.06 | **0.69** |
| HD (1280 x 720) | 0.52 | 0.70 | **0.61** | 0.63 |
| Low (480 x 320) | 0.42 | 0.42 | 0.47 | 0.48 |

Table 7: Higher HD is no longer always better when we fix the the content and vary the appearance.

### E.1 VARIATION INTERACTIONS

To capture possible interactions across variations, we explore combinations of appearance and content changes. We apply combinations of the appearance and content variations across apps to capture their interactions. We then evaluate 12 tasks covering app navigation, adding items to the todo list, and adding calendar events across three agents (GPT-4o vision only, GPT-4o vision + AXTree, and UI-Tars vision only). We report results for these new experiments below highlighting interactions can lead to different success rates (see Table 8). For example, we see UI-Tars and GPT-4o vision agents task success drops when apps contain a combination of English and German compared to German alone.

### E.2 MORE COMPLEX MULTI-STEP TASKS WITH PARTIAL REWARDS

We implement 10 multi-step tasks across todo, calendar, and messenger. These tasks involve for example, looking up an event on the calendar, messaging its title to a friend, and adding a related todo to the todolist. With these multi-step tasks, we also measure incremental rewards if the task was partially completed (say a message was successfully sent, but the todo item wasn't added). We report the reward, capturing incremental rewards as well as a column indicating whether at least one step was successfully completed. As shown in Table 9, we observe agents often struggle to complete multi-step tasks, even GPT-4o with full access to the AXTree. We see for example, GPT-4o and UI-Tars complete single steps towards the task at 3x higher rate than the full multi-step task.

### E.3 VARIATIONS GROUNDED IN POPULAR DESIGN CHOICES AND LANGUAGES USED IN THE WEB

To ground our variations in real world distributions, we run additional experiments using the most popular fonts, colors, and language found on the web. Specifically, we select the three most popular fonts based on data from `https://inkbotdesign.com/popular-fonts/` (Inter, Roboto, Open Sans), three most popular color palettes from `https://colorhunt.co/palettes/popular`, and translate the todo app contents into 3 additional languages based on popularity on the web here: `https://en.wikipedia.org/wiki/Languages_used_on_the_Internet` to cover 4 out of 5 most commonly used languages.

We then evaluate three agents (GPT-4o vision, GPT-4o with vision + AXTree, and UI-Tars vision only UI-model) on three tasks requiring the agent to add items to the todo list. We report these new results grounded in real world variation below as a demonstration of the flexibility of OpenApps in modeling real world variations. The community can of course easily extend these experiments simply by modifying yaml variables for font, button colors, and language via the content.

We report in Table 10 the task success rate for adding items to the todo list across these popular design choices and languages. We find even among these popular choices, task success rates can vary.

## F AGENT BEHAVIORS

**Loops.** In Table 11, we show the average loop count across models, appearance variations, and content variations. When inspecting loops qualitatively, the most common loops are sequences of the same action being repeated, e.g., GPT-4o produces `click(47) click(47) click(47) click(47) click(47) click(47) click(47)`, Kimi-VL generates `click(17) click(17) click(17) click(17) click(17) click(17) click(17) click(17) click(17) click(17) click(17) click(17) click(17) click(17) click(17) click(17) click(17) click(17) click(17)`, and UI-TARS-1.5-7B outputs `keyboard_press(key='ctrl a') keyboard_press(key='ctrl a')`. These examples reveal that loops can be, but are not always problematic. The bid `[17]` corresponds to `Section ''`, which is a task-irrelevant empty section header. In contrast, the bid `[47]`

| agent | variations | task success |
|---|---|---|
| GPT-4o (vision + AxTree) | dark theme | 1.00 |
| | dark theme + default | 1.00 |
| | dark theme + german | 1.00 |
| | default | 1.00 |
| | english + german | 1.00 |
| | german | 1.00 |
| GPT-4o (vision only) | dark theme | 0.11 |
| | dark theme + default | 0.11 |
| | dark theme + german | 0.11 |
| | default | 0.11 |
| | english + german | 0.00 |
| | german | 0.11 |
| UI-TARS-1.5-7B | dark theme | 0.00 |
| | dark theme + default | 0.11 |
| | dark theme + german | 0.11 |
| | English | 0.67 |
| | english + german | 0.67 |
| | german | 0.75 |

Table 8: Variations can interact to form new combinations of app versions. We show agent task success across 15 tasks covering navigation, adding items to todo, and add calendar events.

| agent | variations | reward | at least one correct step |
|---|---|---|---|
| GPT-4o (vision + AxTree) | default | 0.22 | 0.67 |
| GPT-4o (vision only) | default | 0.00 | 0.00 |
| UI-TARS-1.5-7B | default | 0.10 | 0.29 |

Table 9: We show performance on tasks requiring multiple steps across todo, messenger, and calendar. We measure partial reward with respect to whether the agent completed each incremental step in each app. We also show whether at least on step was correctly completed.

| agent | variations | task success |
|---|---|---|
| GPT-4o (vision + AxTree) | Inter | 1.00 |
| | Open Sans | 1.00 |
| | Roboto | 1.00 |
| | default | 1.00 |
| | french | 1.00 |
| | german | 1.00 |
| | popular colors 1 | 1.00 |
| | popular colors 2 | 1.00 |
| | popular colors 3 | 1.00 |
| | spanish | 1.00 |
| GPT-4o (vision only) | Inter | 0.00 |
| | Open Sans | 0.00 |
| | Roboto | 0.00 |
| | default | 0.00 |
| | french | 0.00 |
| | german | 0.00 |
| | popular colors 1 | 0.00 |
| | popular colors 2 | 0.00 |
| | popular colors 3 | 0.00 |
| | spanish | 0.00 |
| UI-TARS-1.5-7B | Inter | 1.00 |
| | Open Sans | 1.00 |
| | Roboto | 0.33 |
| | default | 1.00 |
| | french | 1.00 |
| | german | 1.00 |
| | popular colors 1 | 1.00 |
| | popular colors 2 | 1.00 |
| | popular colors 3 | 1.00 |
| | spanish | 1.00 |

Table 10: We measure agents task success when asked to add items to the todo list across popular choices for fonts, colors, and languages. We find even among the most popular choices, agent success rates can vary.

| Model | Appearance | Content | avg. loop count |
|---|---|---|---|
| claude 4 sonnet | default | german | 0.10 |
| | | long descriptions | 0.11 |
| | dark theme | default | 0.14 |
| | default | adversarial descriptions | 0.14 |
| | challenging font | default | 0.14 |
| | black and white | default | 0.15 |
| | default | misleading descriptions | 0.15 |
| | | default | 0.15 |
| GPT-4o | challenging font | default | 0.32 |
| | black and white | default | 0.34 |
| | dark theme | default | 0.41 |
| | default | default | 0.41 |
| | | misleading descriptions | 0.42 |
| | | adversarial descriptions | 0.44 |
| | | german | 0.45 |
| | | long descriptions | 0.46 |
| Kimi-VL-A3B-Instruct | default | long descriptions | 1.16 |
| | black and white | default | 1.20 |
| | dark theme | default | 1.27 |
| | default | misleading descriptions | 1.33 |
| | challenging font | default | 1.40 |
| | default | default | 1.43 |
| | | german | 1.66 |
| | | adversarial descriptions | 1.74 |
| llava-v1.6-mistral-7b-hf | black and white | default | 1.18 |
| | default | adversarial descriptions | 1.18 |
| | challenging font | default | 1.23 |
| | dark theme | default | 1.24 |
| | default | german | 1.25 |
| | | long descriptions | 1.28 |
| | | misleading descriptions | 1.29 |
| | | default | 1.35 |
| Qwen2.5-VL | default | misleading descriptions | 1.22 |
| | | long descriptions | 1.32 |
| | dark theme | default | 1.33 |
| | challenging font | default | 1.38 |
| | black and white | default | 1.41 |
| | default | default | 1.43 |
| | | adversarial descriptions | 1.45 |
| | | german | 1.80 |
| UI-TARS-1.5-7B | black and white | default | 0.93 |
| | default | german | 1.09 |
| | challenging font | default | 1.21 |
| | default | default | 1.24 |
| | | misleading descriptions | 1.47 |
| | | adversarial descriptions | 1.54 |
| | | long descriptions | 1.63 |
| | dark theme | default | **2.21** |

Table 11: **Agents are more or less prone to loop actions depending on the app variations they encounter.** The average count of loops (across all tasks and random seeds) varies across appearance and content variations. We define loops as maximal sequences of actions entirely consisting of a repeated subsequence. The task prompt is explicit and fixed.

| Model | Appearance | Content | avg. invalid action count |
|---|---|---|---|
| GPT-4o | default | default | 0.00 |
| | dark theme | default | 0.01 |
| | default | adversarial descriptions | 0.07 |
| Kimi-VL-A3B-Instruct | default | default | 0.01 |
| | | german | 0.07 |
| | | long descriptions | 0.07 |
| | black and white | default | 0.07 |
| | default | adversarial descriptions | 0.21 |
| llava-v1.6-mistral-7b-hf | default | german | 0.05 |
| | | misleading descriptions | 0.13 |
| Qwen2.5-VL | default | default | 0.01 |
| | black and white | default | 0.02 |
| | default | misleading descriptions | 0.07 |
| | | german | 0.15 |
| | | long descriptions | 0.20 |
| UI-TARS-1.5-7B | dark theme | default | 0.71 |
| | default | german | 1.84 |
| | black and white | default | 1.88 |
| | challenging font | default | 1.91 |
| | default | default | 2.12 |
| | | misleading descriptions | 2.23 |
| | | long descriptions | 2.25 |
| | | adversarial descriptions | **2.60** |

Table 12: **Agents are more likely to hallucinate actions in certain app variations than others.**
The average count of invalid actions (across all tasks and random seeds) seems to be higher when
the content of apps contains misleading or adversarial descriptions. We define invalid actions as
generated actions that do not have the correct syntax, are not part of the custom or default actions of
a model, or have bad arguments (e.g., type error, argument does not exist). Any unreported models,
appearance variations, and content variations have an average invalid action count of 0. The task
prompt is explicit and fixed.

corresponds to `button 'Next >'` in OpenCalendar, which agents are often required to click
multiple times to navigate to the next calendar page to complete calendar tasks.

**Invalid actions.** Models may hallucinate actions more due to content than appearance variations,
and when an application contains distractor information, e.g., adversarial or misleading descriptions
(see Table 12). The five most common invalid actions for each model include:

- GPT-4o: `click(23)`, `noop`, `check_ax_tree()`, `mouse_click(x=612 y)`

- Kimi-VL-A3B-Instruct: `click(bid)\n'''`, `click(bid)\n<\action>`,
  `click(bid)\n<\action>\n'''python\npyautogui.click(x=0.523,`
  `y=0.466)\n'''`, `click(bid)\n'''\n'''python\npyautogui.click(x=0.000,`
  `y=0.000)\n'''`, `click(bid)\n'''python\npyautogui.click(x=0.000,`
  `y=0.000)\n'''`

- llava-v1.6-mistral-7b-hf: `remove_item(water plants)`, `duplicate(17)`

- Qwen2.5-VL: `click([67] link 'main-page'`, `enter(searchInput,`
  `Bockelwitz, Germany)`, `no-op`, `go_to('https://www.example.com/maps')`,
  `type(OWYN - 100% Vegan Plant-Based Protein Shakes | Cold Brew`
  `Coffee, 12 Fl Oz | Dairy-Free, Gluten-Free, Soy-Free, Tree`
  `Nut-Free, Egg-Free, Allergy-Free, Vegetarian)`

- UI-TARS-1.5-7B: `finished()`, `wait()`, `scroll(direction='down',`
  `point='(966,546)')`, `finished`, `None`

Qualitatively, we observe diverse kinds of invalid actions: incorrectly-formed versions of valid
actions (e.g., `mouse_click(x=612 y)`, `no-op`), well-formed actions that do not exist in

| Model | Appearance | Content | intent misunderstanding rate |
|---|---|---|---|
| GPT-4o | default | german | 0.06 |
| Kimi-VL-A3B-Instruct | default | misleading descriptions | 0.01 |
| | challenging font | default | 0.18 |
| | black and white | default | 0.24 |
| | default | default | 0.25 |
| | dark theme | default | 0.30 |
| | default | german | **0.50** |
| llava-v1.6-mistral-7b-hf | default | misleading descriptions | 0.03 |
| | | adversarial descriptions | 0.06 |
| | | long descriptions | 0.08 |
| | challenging font | default | 0.11 |
| | dark theme | default | 0.12 |
| | black and white | default | 0.18 |
| | default | german | 0.21 |
| | | default | 0.23 |
| Qwen2.5-VL | dark theme | default | 0.02 |
| | default | default | 0.03 |
| | challenging font | default | 0.04 |
| | black and white | default | 0.05 |
| | default | german | 0.31 |
| | | misleading descriptions | 0.36 |
| | | long descriptions | 0.40 |
| | | adversarial descriptions | 0.45 |
| UI-TARS-1.5-7B | challenging font | default | 0.02 |
| | black and white | default | 0.05 |
| | default | default | 0.05 |
| | | german | 0.14 |
| | | adversarial descriptions | 0.31 |
| | | long descriptions | 0.35 |
| | | misleading descriptions | 0.37 |
| | dark theme | default | 0.50 |

Table 13: **Agents may be more prone to misunderstand users' intent when the apps content lengthy, misleading, or adversarial content.** We report the rate of intent misunderstandings over tasks and random seeds for each appearance and content variation. An agent "misunderstands intent" when it navigates to a page irrelevant to the task. Any unreported models, appearance variations, and content variations have an intent misunderstanding rate of 0. The task prompt is explicit and fixed.

the environment (e.g., `remove_item`, `finished`), and valid actions with bad arguments (e.g., `click(bid)`, `scroll(direction='down', point='(966,546)')`).

**Instruction copying.** We do not observe that any models regurgitate in-context examples from their system prompt (e.g., `fill('4', 'my text')`, `mouse_click(200, 300, button='left')`).

**Intent misunderstandings.** In Table 13, we show the intent misunderstanding rate across models, appearance variations, and content variations. We note that we do not consider NavigateToPageTask, as the explicit goal of this task is to navigate to a page rather than to understand user intent. An agent navigating to an incorrect page is not always fatal (i.e., yields an unsuccessful run); however, unnecessary steps in runs can yield higher costs for agent developers. Additionally, agents may navigate to irrelevant pages outside the OPENAPPS environment, e.g., `https://www.example.com/online-shop`, `https://leafletjs.com/`.

