# OpenReview forum: "OpenApps: Simulating Environment Variations to Measure UI Agent Reliability"
_ICLR.cc/2026/Conference — ICLR 2026 Oral_

### Official Review · Reviewer_5Ghj · 2025-10-29

**Soundness:** 2
**Presentation:** 3
**Contribution:** 4
**Rating:** 6
**Confidence:** 4

**Summary:**

This work presents a light-weight version of a benchmark for GUI agents, which presents high reproducibility and easy/cost-effective evaluations. Especially, the authors focused on evaluating the reliability of the agents across UI variants. Across six applications that are popularly employed in many routines, authors configure various combinations of UI diversities, constituting more than 10,000 independent cases. Based on the cases, the vulnerability of agents is revealed, and their qualitative analysis is presented.

**Strengths:**

On top of all, I believe that this work is highly notable because a light-weight benchmark for GUI agents is a missing yet demanding block in this community. I describe several other strengths of this work.
1. Important problem: the reliability of agents in practice is a highly crucial problem. I believe that this work represents an important step forward for the community.
2. Robust success detection logic: the authors put considerable effort into creating success detectors that can work around diverse UI variations. I believe that the efforts that the authors put into this part should be recognized, as this is known to be a challenging problem.
3. Extensive experiments: the volume of the experiments is notable. I believe that such a massive numbers of experiments make the results presented more reliable.
4. Interesting observations: the authors also present a detailed behavioral analysis. These observations highlight the shortcomings of current agents, enabling the design of future solutions.

**Weaknesses:**

I discuss the weaknesses of this paper, including questions and suggestions.
1. Mobile device control benchmarks: while the paper presents fruitful discussion on web benchmarks, discussion on mobile device control benchmarks [1,2,3,4] (which also overlaps with OS-level benchmarks) is absent. Notably, B-MoCA [2] and AndroidWorld [3] both tackle the robustness of the agents, where the former one demonstrates the feature of UI variations and degradation of agents’ performance with respect to the variations, similar to this work. Yet, I do think that this work presents unique and distinct features from prior works (e.g., focus on easy reproduction), which I hope to read in the revision.
2. HTML diversity: while the use of FastHTML is desirable, I question if it would have negative effects in terms of HTML diversity. To elaborate, I think evaluating the agents with diverse formats can be more appealing if the agents take text input.
3. Intentionally misleading description: I worry this is out of scope in this work. There are many works tackling the robustness of the agents in an adversarial manner [5]. However, from my understanding, the UI variations are a ‘natural’ perturbation challenging the agent's robustness rather than ‘intentional’ (line 188-189). Such features should be handled differently, in my opinion.
4. Simplicity of tasks: the proposed test suite suffers from both (1) a lack of diversity and (2) a lack of challenges. There is not enough headroom for improvements in this benchmark for the state-of-the-art agent (i.e., GPT-4o), which I assume would be a bigger problem with the recent agent (e.g., GPT-5). I suggest diversifying the tasks in terms of both volumes (i.e., more than 15 task templates) and difficulties.

I do believe that this work has a strong potential to be a highly noteworthy work that can function as a standardized benchmark.

---

References:

[1] Rawles et al., “Android in the Wild: A Large-Scale Dataset for Android Device Control” (2023).

[2] Lee et al., “Benchmarking Mobile Device Control Agents across Diverse Configurations” (2024).

[3] Rawles et al., “AndroidWorld: A Dynamic Benchmarking Environment for Autonomous Agents” (2024).

[4] Zhang et al., “LlamaTouch: A Faithful and Scalable Testbed for Mobile UI Task Automation” (2024).

[5] Wu et al., “Dissecting Adversarial Robustness of Multimodal LM Agents” (2025).

**Questions:**

For brevity, I included questions and suggestions in the section above.

---

> ### Author Response · Authors · 2025-11-19
>
> We sincerely thank you for thoroughly reading our paper! Here are our responses to address all your concerns:
>
> **Mobile device control benchmarks**
>
> Thank you for referencing these references and acknowledging the unique and distinct features we offer in OpenApps. We agree that these mobile device control benchmarks are relevant to our paper, and we have added the related discussions in the updated Related Work Section. Note that references [1] - [3] above focus uniquely on Android device control, which differs from the objective studied in our paper.
>
> **HTML Diversity**
>
> Thank you for raising this question. We are eager to clarify that the focus of our paper is on agents **that use apps in the same manner as humans, *without* access to HTML**. Thus, instead of providing HTML to agents, in our evaluation, the agent receives observations represented through screenshots and/or an accessibility tree. Compared to HTML, the accessibility tree offers a clean representation of the underlying state, eliminates unnecessary text, and presents only the information that matters for interaction and understanding. Previous work (*e.g.,* (1)) has also noted the brittleness of agents to redundant information in the text input, and thus we argue our choice of accessibility tree is valid.
>
> (1) AgentOccam: A Simple Yet Strong Baseline for LLM-Based Web Agents
>
> **Intentionally misleading description**
>
> Thank you for this question. We agree that a comprehensive evaluation of agents' adversarial robustness falls outside the scope of this paper. However, we want to clarify that we have designed two types of such descriptions. The first type is **misleading description**, which contains information not related to the current goal, and thus could potentially bias the agent to complete another unrelated task. We argue this is a valid *natural* perturbation that challenges the reliability of agents. The second type, closer to your concern and has been tested in previous work, is **adversarial description**, which reflects adversarial or harmful intents against the robustness of agents. It is designed to show promise that our OpenApps *can* be used to study agent adversarial robustness, and we leave broader exploration for this test as future work. We will update the text that describes these two designs to better reflect their functionality and to state the limitations and future work promise more precisely.

---

> > ### Author Response · Authors · 2025-11-19
> >
> > **Simplicity of tasks**
> >
> > Thank you for raising this concern. Based on your feedback, we implement 10 new multi-step tasks across todo, calendar, and messenger. These tasks involve for example, looking up an event on the calendar, messaging its title to a friend, and adding a related todo to the todolist. With these multi-step tasks, we also measure incremental rewards if the task was partially completed (say a message was successfully sent, but the todo item wasn’t added). We report the reward, capturing incremental rewards as well as a column indicating whether at least one step was successfully completed. We see agents often struggle to complete multi-step tasks, even GPT-4o with full access to the AXTree. We see for example, GPT-4o and UI-Tars complete single steps towards the task at 3x higher rate than the full multi-step task. This suggests there is much room for improvement on longer-horizon, multi-step tasks. We thank the reviewer for the suggestion to include these more challenging tasks.
> >
> > | agent                    | variations   |   reward |   at_least_one_step_correct |
> > |:-------------------------|:-------------|---------:|----------------------------:|
> > | GPT-4o (vision + AxTree) | default      |     0.22 |                        0.67 |
> > | GPT-4o (vision only)     | default      |     0.00 |                        0.00 |
> > | UI-TARS-1.5-7B           | default      |     0.10 |                        0.29 |
> >
> >
> > Furthermore, as discussed in the introduction and Section 3.3, we anticipate rapid advancements in the field of autonomous agents due to its early stage of development. Consequently, we would like to highlight that we have designed OpenApps beyond a traditional benchmark. It is *extensible*, allowing researchers to easily configure their own tasks and variations with varying complexity.
> >
> > Finally, we expand the coverage of our test by running additional experiments using the most popular fonts, colors, and language found on the web. Specifically, we select the three most popular fonts based on data from https://inkbotdesign.com/popular-fonts/ (Inter Roboto Open Sans), three most popular color palettes from https://colorhunt.co/palettes/popular, and translate the app contents into 3 additional languages based on popularity on the web here: https://en.wikipedia.org/wiki/Languages_used_on_the_Internet to cover 4 out of 5 most commonly used languages. We then evaluate UI-Tars, a multimodal agent relying on visual inputs on three tasks requiring the agent to add items to the todo list. We report these new results grounded in real world variation below as a demonstration of the flexibility of OpenApps in modeling real world variations. As mentioned above, the community can of course easily extend these experiments simply by modifying yaml variables for font, button_colors, and language via the content.
> >
> > We report in Section E.3 (and table below) the task success rate for adding items to the todo list  across these popular design choices and languages.  We find UI-Tars is robust to the three popular colors palettes and languages. In contrast, UI-Tars is affected by choices of fonts, highlighting the importance of measuring reliability across app variations even among popular choices of fonts.
> >
> > | agent                    | variations       |   task success |
> > |:-------------------------|:-----------------|---------------:|
> > | UI-TARS-1.5-7B           | Inter            |           1.00 |
> > | UI-TARS-1.5-7B           | Open Sans        |           1.00 |
> > | UI-TARS-1.5-7B           | Roboto           |           0.33 |
> > | UI-TARS-1.5-7B           | popular colors 1 |           1.00 |
> > | UI-TARS-1.5-7B           | popular colors 2 |           1.00 |
> > | UI-TARS-1.5-7B           | popular colors 3 |           1.00 |
> > | UI-TARS-1.5-7B           | german           |           1.00 |
> > | UI-TARS-1.5-7B           | english          |           1.00 |
> > | UI-TARS-1.5-7B           | french           |           1.00 |
> > | UI-TARS-1.5-7B           | spanish          |           1.00 |

---

> > > ### Comment · Reviewer_5Ghj · 2025-11-23
> > > **Comments on the rebuttal**
> > >
> > > Thanks for clarifying and reflecting on my reviews. I remain positive on this paper and raised the score from 6 to 8. Here are some comments on your rebuttal.
> > > 1. Mobile device control benchmarks: I am grateful if this helped improve your work.
> > > 2. HTML Diversity: Thanks for the clarification. Why is using accessibility tree evaluating agents using apps in the same manner as humans?
> > > 3. Intentionally misleading description: I agree that the natural perturbation can be a realistic challenge to the agents. I also believe that leaving OpenApps usable for the second type is a reasonable point.
> > > 4. Simplicity of tasks: I agree that the tasks already presented are fundamental and important, while I am still concerned about the simplicity, considering the rapid improvement of this community. I value the additional experimental results. That said, I suggest including discussions of possible future extension, especially which apps/tasks to include, in the revision. (+ Just for curiosity, can some apps regarding “video edit”, “settings (e.g., display setting, os version check)”, “stock trading” be easily incorporated in your benchmark? I think including this specific question/discussion in the paper is not necessary.)

---

### Official Review · Reviewer_1P4o · 2025-10-31

**Soundness:** 3
**Presentation:** 3
**Contribution:** 3
**Rating:** 6
**Confidence:** 4

**Summary:**

This paper introduces OpenApps, a novel, lightweight, open-source ecosystem designed to measure a new dimension of reliability for autonomous UI-agents: performance fluctuations across app variations. Current evaluations rely on fixed environments, failing to capture how changes in app design, appearance, or content affect agent success. OPENAPPS provides six common apps (calendar, maps, messenger, etc.) that are highly configurable via simple Python/YAML files, enabling large-scale, reproducible experiments (over 10,000 trials conducted). The study finds that while reliability within a fixed app is stable, reliability across app variations fluctuates drastically (e.g., Kimi-VL-3B's success rate varied from 63% to 4%), and that failure modes like looping and hallucination are highly environment-dependent. This highlights app variation as a critical, overlooked axis of agent reliability.

**Strengths:**

1. The paper correctly identifies the critical gap between testing on fixed clones and real-world deployment, where app style, content density, and language constantly change. This significantly advances the utility of reliability metrics.

2. OpenApps is designed for parallel, large-scale study (single CPU, < 10MB memory, pure Python). This low overhead is a crucial enabler for large-scale RL training and comprehensive testing that previous VM-based or complex web-clone environments could not afford.

3. Beyond average success rates, the study provides valuable diagnostics on how environment changes induce specific failure modes.

**Weaknesses:**

1. The paper focuses exclusively on 15 simple, short-horizon tasks (e.g., adding a to-do item). While justified to isolate reliability, the performance of agents on complex, multi-app workflows (which is the ultimate promise of UI-agents) remains unevaluated. Future work needs to extend the task set to long-horizon, multi-step tasks to form a comprehensive reliability benchmark.
2. The study primarily varies each app factor (appearance or content) independently. The authors acknowledge that interactions between multiple variations (e.g., dark theme + German language + dense content) could expose novel failure modes. A brief initial exploration of combined variations would strengthen the argument.
3. The paper provides excellent diagnostics but offers limited concrete proposals for agent development (e.g., does VLM fine-tuning on synthetic variants solve the problem? Should contrast-boosting pre-processing be used for dark themes?). While the goal is diagnosis, suggesting a simple architecture or training technique that could leverage the synthetic data would enhance the paper's prescriptive value.

**Questions:**

1. The flexibility of OpenApps allows generating training data across thousands of versions. Did the authors explore whether fine-tuning a model (e.g., UI-TARS) on a synthetic dataset generated by OpenApps could significantly improve its overall deviation (Figure 5) compared to training only on a single fixed version?

2. The paper notes GPT-4o performs highly when given simplified AX tree representations along with the screenshot. Which specific content variations (e.g., German language, adversarial text) affected the reliability of AX tree parsing more than the visual recognition, or was the failure primarily a model-level reasoning issue?

3. The reward function is deterministic. For complex, open-ended tasks, a continuous partial reward is often more useful. Could the authors confirm that the Python logic of OpenApps could support easily pluggable, continuous reward functions (e.g., L2 distance between the current state vector and the target state vector) for future RL training efforts?

---

> ### Author Response · Authors · 2025-11-19
>
> We thank Reviewer 1P4o for the thoughtful and constructive feedback. We address your questions and concerns below:
>
> **Responses to Questions**
>
> 1. Fine-tuning on Synthetic Data:
> Thank you for this excellent suggestion. While our current work focuses on evaluation rather than training, we agree that leveraging OpenApps to generate synthetic data for fine-tuning is a promising direction. We plan to explore this in future work and believe OpenApps provides a strong foundation for such experiments.
>
> 2. AX Tree vs. Visual Recognition:
> We appreciate your interest in the diagnostic details. As shown in Table 5 and Table 6, we compared the effects of content and appearance variations. From working with other agent benchmarks, we observed that AX tree representations sometimes fail to capture all necessary information, particularly due to misidentification of clickable elements. To mitigate this, we ensured that, for OpenApps, AX trees were sufficient and consistent across appearance variations.
>
> 3. Support for Continuous Reward Functions:
> Yes, OpenApps is designed to be highly configurable. The Python logic allows for easily pluggable, continuous reward functions, such as Manhattan distance between state vectors. We have added clarifications and examples in Section E.2 of the revised manuscript to make this explicit. Please also see the general response.
>
> **Responses to Criticisms**
>
> 1. Task Complexity:
> We agree that evaluating agents on more complex, long-horizon, and multi-app tasks is an important next step. Our current focus on short-horizon tasks is motivated by the need to isolate and understand reliability issues. Notably, agents still struggle with these simpler tasks, highlighting the challenge. As agent capabilities improve, our framework is well-suited to support more complex benchmarks in future work. Based on your feedback, we implement 10 new multi-step tasks across todo, calendar, and messenger. These tasks involve for example, looking up an event on the calendar, messaging its title to a friend, and adding a related todo to the todolist. With these multi-step tasks, we also measure incremental rewards if the task was partially completed (say a message was successfully sent, but the todo item wasn’t added). We report the reward, capturing incremental rewards as well as a column indicating whether at least one step was successfully completed. We see agents often struggle to complete multi-step tasks, even GPT-4o with full access to the AXTree. We see for example, GPT-4o and UI-Tars complete single steps towards the task at 3x higher rate than the full multi-step task. This suggests there is much room for improvement on longer-horizon, multi-step tasks. We thank the reviewer for the suggestion to include these more challenging tasks.
>
>
> | agent                    | variations   |   reward |   at_least_one_step_correct |
> |:-------------------------|:-------------|---------:|----------------------------:|
> | GPT-4o (vision + AxTree) | default      |     0.22 |                        0.67 |
> | GPT-4o (vision only)     | default      |     0.00 |                        0.00 |
> | UI-TARS-1.5-7B           | default      |     0.10 |                        0.29 |

---

> > ### Author Response · Authors · 2025-11-19
> >
> > 2. Combined Variations:
> > Thank you for raising this point. In response, we have conducted additional experiments exploring variation combination interactions. To capture possible interactions across variations, we explore combinations of appearance and content changes. We apply combinations of the appearance and content variations across apps to capture their interactions. We then evaluate 12 tasks covering app navigation, adding items to the todo list, and adding calendar events across two visual agents (GPT-4o vision and UI-Tars). We report results for these new experiments below highlighting interactions between variations can lead to new app versions with distinct success rates; for example combinations of German and English or Dark and Default Themes mixed have different success rates than individual variations. This suggests OpenApps can be scaled not only by varying content and appearance of apps, but also by combining variations. We thank the reviewer for the suggestion to study interactions and include these new results in our updated manuscript in Section E.1.
> >
> > | agent                    | variations           |   task success |
> > |:-------------------------|:---------------------|---------------:|
> > | GPT-4o (vision only)     | dark theme           |           0.11 |
> > | GPT-4o (vision only)     | dark theme + default |           0.11 |
> > | GPT-4o (vision only)     | dark theme + german  |           0.11 |
> > | GPT-4o (vision only)     | default              |           0.11 |
> > | GPT-4o (vision only)     | english + german     |           0.00 |
> > | GPT-4o (vision only)     | german               |           0.11 |
> > | UI-TARS-1.5-7B           | dark theme           |           0.00 |
> > | UI-TARS-1.5-7B           | dark theme + default |           0.11 |
> > | UI-TARS-1.5-7B           | dark theme + german  |           0.11 |
> > | UI-TARS-1.5-7B           | default              |           0.67 |
> > | UI-TARS-1.5-7B           | english + german     |           0.67 |
> > | UI-TARS-1.5-7B           | german               |           0.75 |
> > 3. Prescriptive Value for Agent Development:
> > We appreciate this suggestion. While our primary goal is diagnostic, we agree that providing concrete proposals for leveraging synthetic data would enhance the paper. We include initial ideas in Section B covering future work.
> >
> >
> > We thank the reviewer again for the insightful feedback and hope our revisions address your concerns. We welcome any further suggestions or questions.

---

### Official Review · Reviewer_CTPq · 2025-11-02

**Soundness:** 4
**Presentation:** 3
**Contribution:** 4
**Rating:** 8
**Confidence:** 4

**Summary:**

This paper introduces OpenApps, a lightweight and scalable benchmark designed to evaluate the **reliability of multimodal UI agents under app-environment variations**. It exposes an overlooked weakness in current evaluation protocols—agents’ fragility to minor visual or structural changes in otherwise identical tasks. The framework is technically sound, reproducible, and empirically comprehensive across seven models. Results show dramatic reliability drops (up to 50% across app versions) and reveal new failure behaviors such as looping and hallucination. The contribution is both **timely and impactful**, offering a reproducible infrastructure that can underpin future robustness research.

While methodologically solid, the work would benefit from a **clearer theoretical framing of “reliability,” richer task diversity, and improved presentation**. The current tasks are simple and limited to single-app workflows; including long-horizon tasks would strengthen generality. Despite these limitations, the paper’s novelty, execution quality, and potential community value are strong.

**Strengths:**

1. The paper identifies a key blind spot in existing agent benchmarks: reliability across app variations. Unlike prior environments that focus on fixed app clones, OpenApps systematically quantifies how changes in design and content affect UI-agent performance. This is a genuine conceptual contribution to the field of multimodal agent evaluation.
2. OpenApps is implemented in pure Python and runs on a single CPU, removing the heavy dependencies of prior environments (e.g., emulators, Docker containers, or large memory requirements). This design choice makes it widely accessible and reproducible.
3. The authors conduct 10,000+ trials across seven state-of-the-art agents, including GPT-4o, Claude Sonnet, Kimi-VL, Qwen-VL, and UI-TARS. The scale and comprehensiveness of these experiments convincingly demonstrate the practical significance of app variation as a factor in agent reliability.
4. Results are striking: task success can fluctuate by more than 50% across app versions, and specific models show massive degradation (e.g., Kimi-VL drops from 63% to 4%). The paper also documents behavioral shifts such as looping and hallucination, revealing that environmental variability induces new failure modes.
5. OpenApps can be extended for safe training, adversarial robustness testing, or sim2real transfer studies. The discussion section articulates a credible roadmap for future research directions, showing awareness of the broader ecosystem.

**Weaknesses:**

1. **Task Simplicity and Limited Scope**
    - The current experiments focus on **15 simple tasks** (e.g., adding a to-do item). While the paper demonstrates large variability even on these, such simplicity limits conclusions about generalization to *complex or long-horizon* tasks seen in real-world apps.
2. **Insufficient Theoretical Framing of “Reliability”**
    - While empirical results are strong, the paper lacks a deeper theoretical formalization of *reliability across app variations*—for example, framing it as an expected reward stability problem under distributional shifts could strengthen the conceptual rigor.

**Questions:**

+ How are app variations generated — random parameter changes, manually curated modifications, or rule-based templates? Could the process introduce artificial correlations that models might exploit?
+ How do you ensure that a task instance in version A is semantically equivalent to the same task in version B (e.g., identical goal, only UI difference)?
+ Did you observe any model types that adapt better to variations (e.g., multimodal LLMs vs. RLVR UI-trained agents)? If so, what might explain this?

---

> ### Author Response · Authors · 2025-11-19
>
> We very much appreciate your careful read of our work and helpful suggestions. We’re very glad to see you appreciated the core contribution of our work in evaluating the overlooked dimension of reliability across app variations. We agree the findings are striking in highlighting the importance of this overlooked dimension of reliability. We’re also thrilled you noted the effort we made in building OpenApps as a lightweight framework written in Python with reproducibility in mind.
>
> **New More Complex Multi-Step Tasks**
>
> We agree our initial seed of tasks is based on simple tasks requiring just a few tasks. While we found these simple tasks sufficient to study agent task success fluctuations across app variations, we agree our work could benefit from longer-horizon more complex tasks. Based on your feedback, we implement 10 new multi-step tasks across todo, calendar, and messenger. These tasks involve for example, looking up an event on the calendar, messaging its title to a friend, and adding a related todo to the todolist. With these multi-step tasks, we also measure incremental rewards if the task was partially completed (say a message was successfully sent, but the todo item wasn’t added). We report the reward, capturing incremental rewards as well as a column indicating whether at least one step was successfully completed. We see agents often struggle to complete multi-step tasks, even GPT-4o with full access to the AXTree. We see for example, GPT-4o and UI-Tars complete single steps towards the task at 3x higher rate than the full multi-step task. This suggests there is much room for improvement on longer-horizon, multi-step tasks. We thank the reviewer for the suggestion to include these more challenging tasks.
>
> | agent                    | variations   |   reward |   at_least_one_step_correct |
> |:-------------------------|:-------------|---------:|----------------------------:|
> | GPT-4o (vision + AxTree) | default      |     0.22 |                        0.67 |
> | GPT-4o (vision only)     | default      |     0.00 |                        0.00 |
> | UI-TARS-1.5-7B           | default      |     0.10 |                        0.29 |
>
>
> As discussed in the introduction and Section 3.3, we anticipate rapid advancements in the field of autonomous agents due to its early stage of development. Consequently, we would like to highlight that we have designed OpenApps beyond a traditional benchmark. It is *extensible*, allowing researchers to easily configure their own tasks and variations with varying complexity.
>
> **Theoretical Framing of “Reliability”**
>
> Thank you for the suggestion. While we favored accessibility in our initial manuscript, we agree reliability across apps can be cast as a form of distribution shift to better ground our work in existing theoretical frameworks. We have added a new paragraph Section 4 highlighting how each app variation can be seen as an environment representing a distribution shift. To measure reliability we compare the fluctuation in reward within an observable environment to the fluctuation in task success across distribution shifts, which we highlight can be considerable.
>
> **Questions**
>
> 1. Each app has a set of configurable factors that can be specified via a yaml file. We then manually curate an initial set of variation as described in section XX. Since we evaluate agents without finetuning on OpenApps agents do not have the chance to exploit correlations or shortcuts specific to OpenApps. In future work however, we agree it’s important to keep in mind precisely how training and evaluation splits are crafted to avoid such correlation exploits. We’ve updated Section B of the Appendix to specifically call out this important point.
> 2. We ensure each task instance is identical by ensuring each task instance across runs is identical (same goal, reward function, and allotted number of steps). Practically, each task is a Python class with arguments that are specified via yaml files that allow us to precisely specify the goal, reward function to ensure each task instance is identical.
> 3. We found UI-Tars, a specialized multimodal UI agent to be most robust across multimodal agents we explored. In contrast, general purpose multimodal language models, including reasoning based Kimi-VL models, tended to have larger fluctuations in task success. This suggests domain specific UI training, which exposes the foundation model to a richer set of variations, is a promising path to more reliable UI agents. The configurability of OpenApps is a promising first step towards such domain-specific training.

---

### Official Review · Reviewer_MLr5 · 2025-11-02

**Soundness:** 3
**Presentation:** 4
**Contribution:** 3
**Rating:** 6
**Confidence:** 4

**Summary:**

The paper addresses the question of reliability of UI agents when the underlying environments change (without affecting overall functionality). The key idea is to create multiple appearance and content variations of a set of UI apps and measure agent performance across these variations. The authors find that all models have significant variance in performance, with larger models being relatively more stable compared to smaller models. The analysis further illustrates that certain specific types of variations in the environment lead to consistently larger drops in agent performance (e.g. using darker themes for apps leads to consistent failures).

**Strengths:**

- The question of reliability is an important one, especially when considering how dynamic real world apps often are. Assessing agent performance under these variations is quite useful.

- The analysis shows that multimodal models have large variability in performance, raising important questions about their suitability in practical settings, where the underlying environments change.

- This benchmark could be a useful additional evaluation for multimodal agent solutions in addition to the existing benchmarks.

**Weaknesses:**

The main issue I have with this work is that the motivations for the variations is not adequately justified and the choices for curation are not well explained.
  - Content variations seem somewhat arbitrary. If one were to include misleading descriptions and adversarial perturbations performance is going to drop. What is the point of this exercise? There is not much in terms of motivation for why these were done and what specific ways in which these were created. Why use German translations? Why not others?
  - Similarly the choice of stylistic variations also appears somewhat arbitrary and adversarial. It would have been much more natural to target the most frequently used variances alongside some of the more rarely used variations.

**Questions:**

- I appreciate the state based evaluation and the rigorous evaluation. However, it seems like partial progress towards the task is not accounted for in the current analyses. Is there a way to extend the analysis to partial progress?

- How are the within app fluctuations observed? Are these deviations computed across all queries? Or averages of standard deviations over multiple attempts for each query?

---

> ### Author Response · Authors · 2025-11-19
>
> We appreciate your careful review and suggestions and are especially glad to see you recognize the importance of reliability for agents deployed in real world apps.  We also absolutely agree OpenApps can serve the community as an additional benchmark to capture this missing dimension of reliability for multimodal agents.
>
> **New Variations Motivated by Real World Distributions**
>
> We agree with you the variations we include could use better grounding in real world distributions. Based on your suggestion, we including several new variations in OpenApps mirroring the most popular choices used in real world sites. Specifically, we
>
> To capture possible interactions across variations, we explore combinations of appearance and content changes. We apply combinations of the appearance and content variations across apps to capture their interactions. We then evaluate 12 tasks covering app navigation, adding items to the todo list, and adding calendar events across two visual agents (GPT-4o vision and UI-Tars). We report results for these new experiments below highlighting interactions between variations can lead to new app versions with distinct success rates; for example combinations of German and English or Dark and Default Themes mixed have different success rates than individual variations. This suggests OpenApps can be scaled not only by varying content and appearance of apps, but also by combining variations. We thank the reviewer for the suggestion to study interactions and include these new results in our updated manuscript in Section E.1.
>
> | agent                    | variations           |   task success |
> |:-------------------------|:---------------------|---------------:|
> | GPT-4o (vision only)     | dark theme           |           0.11 |
> | GPT-4o (vision only)     | dark theme + default |           0.11 |
> | GPT-4o (vision only)     | dark theme + german  |           0.11 |
> | GPT-4o (vision only)     | default              |           0.11 |
> | GPT-4o (vision only)     | english + german     |           0.00 |
> | GPT-4o (vision only)     | german               |           0.11 |
> | UI-TARS-1.5-7B           | dark theme           |           0.00 |
> | UI-TARS-1.5-7B           | dark theme + default |           0.11 |
> | UI-TARS-1.5-7B           | dark theme + german  |           0.11 |
> | UI-TARS-1.5-7B           | default              |           0.67 |
> | UI-TARS-1.5-7B           | english + german     |           0.67 |
> | UI-TARS-1.5-7B           | german               |           0.75 |

---

> > ### Author Response · Authors · 2025-11-19
> >
> > **Rewards for partial progress**
> >
> > We agree, this is an excellent suggestion. We can easily implement partial rewards by rewarding partial state shifts. Specifically, instead of only checking if the final target state is reached, we can reward the agent for achieving intermediate parts of that state.
> > To demonstrate our point empirically, we implement ten new multi-step tasks where measuring partial rewards is important (because existing tasks require only one or two steps). We implement these new tasks and measure partial reward if only a subset of the steps needed are complete.
> >
> >
> > We implement 10 new multi-step tasks across todo, calendar, and messenger. These tasks involve for example, looking up an event on the calendar, messaging its title to a friend, and adding a related todo to the todolist. With these multi-step tasks, we also measure incremental rewards if the task was partially completed (say a message was successfully sent, but the todo item wasn’t added). We report the reward, capturing incremental rewards as well as a column indicating whether at least one step was successfully completed. We see agents often struggle to complete multi-step tasks, even GPT-4o with full access to the AXTree. We see for example, GPT-4o and UI-Tars complete single steps towards the task at 3x higher rate than the full multi-step task. This suggests there is much room for improvement on longer-horizon, multi-step tasks. We thank the reviewer for the suggestion to include these more challenging tasks.
> >
> > | agent                    | variations   |   reward |   at_least_one_step_correct |
> > |:-------------------------|:-------------|---------:|----------------------------:|
> > | GPT-4o (vision + AxTree) | default      |     0.22 |                        0.67 |
> > | GPT-4o (vision only)     | default      |     0.00 |                        0.00 |
> > | UI-TARS-1.5-7B           | default      |     0.10 |                        0.29 |
> >
> > **How are within app fluctuations observed?**
> >
> > That’s right, within app fluctuations measures task success fluctuations when an agent is given multiple attempts for a task. Specifically, to measure within app fluctuations, we compute the standard deviation of reward over multiple attempts for the same task. Each task has multiple goal statements and agents have multiple attempts at each goal statement per task. The intent is to capture the standard fluctuations in task success a researcher would be able capture with a static benchmark. OpenApps complements this by also capturing the task success fluctuations across the distributions of app variations agents would encounter. As shown in Figure 5, we see within app fluctuations in task success underestimate the larger fluctuations in task reward agents would encounter across app variations.
> >
> > We thank you again for these helpful suggestions and hope the new experiments based on your feedback addressed your concerns. Please know we remain available to answer any questions or further discussion.

---

### Author Response · Authors · 2025-11-19

We’re grateful to the reviewers for their careful consideration of our work. We’re thrilled to see reviewers appreciated the

- **Importance of agent reliability across app variations**: reviewers commented on the “highly crucial problem” [5Ghj] we study, noting “The question of reliability is an important one, especially when considering how dynamic real world apps often are ” [MLr5]. Compared to existing work reviewers noted “the paper identifies a key blind spot in existing agent benchmarks” [CTPq]  “significantly advances the utility of reliability metrics” [1P4o].
- **Low overhead of OpenApps making large scale agent research accessible**: reviewers mentioned our work is “highly notable because a light-weight benchmark,” calling out “OpenApps is implemented in pure Python and runs on a single CPU, removing the heavy dependencies of prior environments” [CTPq]. Reviewers noted the importance of these design choices in terms of scale:  “this low overhead is a crucial enabler for large-scale RL training and comprehensive testing that previous VM-based or complex web-clone environments could not afford” [1P4o]
- **Systematic experiments with striking findings**: “authors conduct 10,000+ trials across seven state-of-the-art agents” [CTPq] and that “the volume of the experiments is notable”[5Ghj].  Reviewers characterized the findings as “Results are striking: task success can fluctuate by more than 50% across app versions” [CTPq] “valuable diagnostics on how environment changes induce specific failure modes.” [1P4o] “this work has a strong potential to be a highly noteworthy work that can function as a standardized benchmark.” [5Ghj].

Reviewers made several suggestions regarding the complexity of tasks, app variation coverage, and presentation of our work. Thanks to their suggestions, we’ve made a considerable effort to revise our manuscript based on their feedback and run three new experiments:

1. **New more complex, multi-step tasks with partial rewards**: three reviewers, 1P4o, n5Ghj, CTPq suggested we consider more complex long-horizon tasks. Based on their feedback, we implemented 10 new multi-step tasks (see Section E.2). For these complex tasks we also incorporate reviewers’ MLr5 and 1P4o feedback to include partial rewards. We find multi-step tasks to be more challenging for agents.
2. **New experiments covering popular app variation font, color, and language choices**: based on reviewers MLr5 and CTPq’s suggestions, these experiments extend our coverage of app variations to popular choices found on the web (see Section E.3)
3. **New experiments studying interactions across app variations**: based on reviewer 1P4o’s suggestion, these experiments highlight OpenApps scale to generate even more app versions by considering combinations of app variations (see Section E.1).

We’ve also updated our manuscript with a theoretical grounding of reliability across app variations in terms of distribution shifts in Section 4. All revisions are highlighted in blue to ease reviewers’ burden. We once again thank reviewers for their outstanding suggestions that we believe have greatly improved the quality of our work. With reviewers’ feedback incorporated, we believe OpenApps can serve as a foundation to the community for accessible multimodal agent reliability research.

---

### Meta-Review · Area_Chair_HgHR · 2025-12-14

**Summary:**

OpenApps is a timely, lightweight, and scalable benchmark that addresses a critical gap in the evaluation of multimodal UI agents. By simulating diverse changes in appearance, content, and structure across common UI applications, OpenApps enables rigorous, reproducible assessments at scale. The authors conducted over 10,000 trials across seven state-of-the-art agents, uncovering striking performance degradations due to UI changes. With its minimal resource requirements and high configurability, OpenApps offers a practical, extensible platform for studying robustness, failure modes, and distribution shifts in agent behavior. It shows potential to become a community-standard benchmark for large-scale, real-world reliability evaluation.

**Reviewer Concerns:**

Reviewers 1P4o, 5Ghj, and CTPq requested more complex tasks. In response, the authors introduced 10 new multi-step tasks with partial rewards, demonstrating that current agents struggle even with incremental progression.

Reviewers MLr5 and 5Ghj raised concerns about the arbitrariness of UI variations. The authors addressed this by conducting new experiments using widely adopted fonts (e.g., Inter, Roboto, Open Sans), standard color palettes, and multiple languages (English, German, French, Spanish), grounding the variations in real-world usage.

Reviewer 1P4o suggested exploring the interaction effects of multiple variations. The authors responded with new experiments highlighting compositional failures, such as performance degradation under the combination of a dark theme and the German language.

**Reviewer Scores:**

Reviewer 5Ghj explicitly raised their score from 6 to 8. The remaining reviewers, who were initially at 8 and 6, would likely maintain or even increase their scores in light of the thorough and substantive rebuttal.

---

### Decision · Program_Chairs · 2026-01-26

Accept (Oral)